# See What Matters: Differentiable Grid Sample Pruning for Generalizable Vision-Language-Action Model

**Yixu Feng** [1]  **Zinan Zhao** [2] [†]  **Yanxiang Ma** [1]  **Chenghao Xia** [3]  **Chengbin Du** [3]  **Yunke Wang** [1]  **Chang Xu** [1]

## Abstract

Vision-Language-Action (VLA) models have shown remarkable promise in robotics manipulation, yet their high computational cost hinders real-time deployment. Existing token pruning methods suffer from a fundamental trade-off: aggressive compression using pruning inevitably discards critical geometric details like contact points, leading to severe performance degradation. This forces a compromise, limiting the achievable compression rate and thus the potential speedup. We argue that breaking this trade-off requires rethinking compression as a geometry-aware, continuous token resampling in the vision encoder. To this end, we propose the *Differentiable Grid Sampler (GridS)*, a plug-and-play module that performs task-aware, continuous resampling of visual tokens in VLA. By adaptively predicting a minimal set of salient coordinates and extracting features via differentiable interpolation, GridS preserves essential spatial information while achieving drastic compression (with fewer than 10% original visual tokens). Experiments on both LIBERO benchmark and a real robotic platform demonstrate that validating the lowest feasible visual token count reported to date, GridS achieves a 76% reduction in FLOPs with no degradation in the success rate. The code is available at Github.

## 1. Introduction

Vision-Language-Action models (VLAs) integrate semantic reasoning and low-level control into unified architectures, showing strong generalization across manipulation tasks (Bai et al., 2025; Black et al., 2024; Kim et al., 2025b; Zitkovich et al., 2023; Zhong et al., 2026; Pertsch et al.,

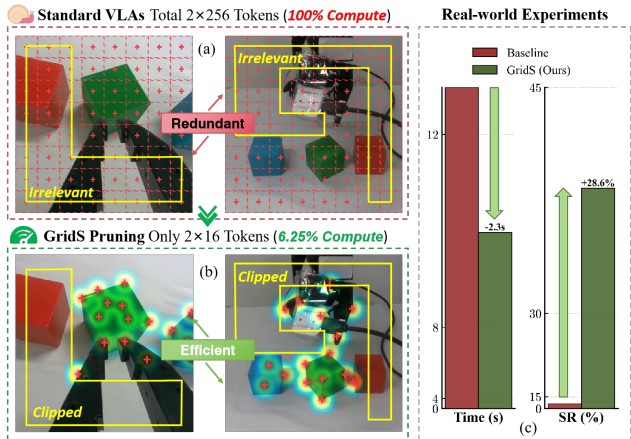

*Figure 1.* Motivation and Performance of GridS. (a) Standard VLAs process images with dense, uniform token representations ($2 \times 256$), leading to high computational redundancy in irrelevant background areas (100% Compute). (b) Our Grid Sample (GridS) prunes non-essential tokens, focusing only on salient regions. This reduces the token count to $2 \times 16$, requiring only 6.25% of the original compute. (c) Real-world Experiments demonstrate that GridS not only reduces inference latency by 2.3s but also significantly boosts the out-of-distribution task success rate (SR) by $+28.6\%$ compared to the baseline.

2025; Liang et al., 2025; Xu et al., 2025a;c; Hui et al., 2026; Li et al., 2026a). However, VLAs incur high computational costs due to the massive token volume of the vision encoder. As shown in Fig. 1(a), an image is projected into 256 tokens in vision encoder (Dosovitskiy et al., 2021). This excessive sequence length creates a severe bottleneck for the subsequent Transformer block, where computational complexity scales quadratically with input sequence length.

To mitigate this efficiency bottleneck, recent research has investigated visual token pruning. One prevalent approach adopt semantic pruning in VLAs (Li et al., 2026c), yet these methods often prioritize salient object bodies while neglecting geometrically critical but semantically subtle regions, *e.g.* edges or contact points (Qu et al., 2025; Chen et al., 2024a). Another direction is dynamic token pruning, which retains variable token counts per image (Yang et al., 2025; Xu et al., 2025b; Zhang et al., 2025; Xu et al., 2026; Pei et al., 2026). These methods adopt training-free strategies

[†] Work done during internship at StellarEdge Robotics. [1] The University of Sydney [2] City University of Hong Kong [3] StellarEdge Robotics. Correspondence to: Chang Xu <c.xu@sydney.edu.au>.

*Proceedings of the $43^{rd}$ International Conference on Machine Learning*, Seoul, South Korea. PMLR 306, 2026. Copyright 2026 by the author(s).

that are inherently non-differentiable, functioning as static heuristics derived from pre-existing policies. Consequently, they fail to align with specific application requirements, leading to varying degrees of degradation in task success rates.

More fundamentally, both types of methods are constrained by their reliance on discrete selection over a fixed grid (Yuan et al., 2025). However, robotic interaction often requires sub-patch precision (as Fig. 2 (a), a grasp point is located between two patches). Restricting the model to select coarse patches introduces inherent quantization errors and leads to a loss of fine-grained spatial fidelity.

To break this performance-efficiency trade-off, we propose the Differentiable **Grid S**ampler (**GridS**). GridS fundamentally reformulates VLA token compression from a passive discrete patch-dropping task into an active, geometry-aware continuous resampling process. Rather than being confined to the rigid output grid of a vision encoder, GridS dynamically predicts task-driven spatial coordinates and queries the continuous feature map via differentiable bilinear interpolation. As shown in Figs 1 and 2 (b), our method precisely localizes and samples the most discriminative tokens, retaining only 6.25% —(or even lower) of the original count. This allows the model to filter out a few task-related regions from the dense tokens, serving as an effective pruning strategy.

We evaluate GridS in two flow-matching VLA architectures ($\pi_0$ (Black et al., 2024), $\pi_{0.5}$ (Intelligence et al., 2025)), and a auto-regressive model (SmolVLA (Shukor et al., 2025)) to demonstrate its superior efficiency. In simulation tasks (LIBERO (Liu et al., 2023) and ALOHA (Zhao et al., 2023)), GridS maintains baseline success rates using fewer than 10% of visual tokens, while increasing task speed by $1.2\times$ and training speed by $3.4\times$. In particular, on the LIBERO task of $\pi_0$, we achieved a 1% improvement in accuracy by only using 4 visual tokens. Real-world experiments on a consumer-grade GPU reveal an average $1.23\times$ inference speedup and a 22.2% success rate gain over baseline (SmolVLA). Counter-intuitively, we observe that reducing the number of tokens leads to a substantial 28.6% accuracy gain in out-of-distribution (OOD) scenarios (see Fig. 1 (c)). By significantly reducing deployment costs, GridS establishes a new pruning paradigm, paving the way for VLAs in highly dynamic and physically interactive environments. Furthermore, guided by constructive reviewers feedback, we have expanded our evaluations to include the LIBERO-PLUS (Fei et al., 2025) and RoboTwin (Mu et al., 2025) benchmarks, detailed in App. E. Crucially, these extended stress tests highlight the ultimate potential of our compression paradigm: GridS can aggressively distill visual information down to a single token ($K = 1$) while maintaining on-par performance with the dense baseline (reported in App. G).

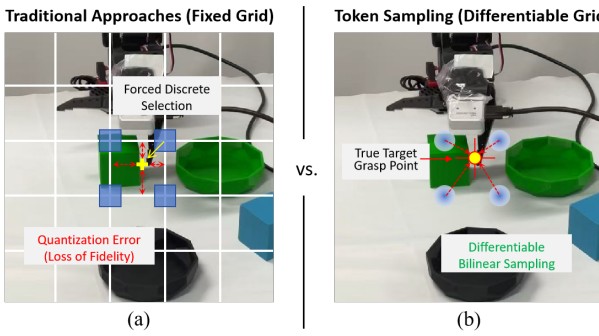

*Figure 2.* Discrete Selection vs. Differentiable Sampling. (a) Traditional approaches operate on a fixed grid. When the target region (yellow cross) falls between patches, the model is forced to perform discrete selection, leading to spatial quantization errors and a loss of fidelity. (b) Our approach predicts continuous coordinates and utilizes differentiable bilinear sampling to interpolate features from the four nearest neighbors.

**Conflict of Interest Disclosure.** The authors declare that they have no financial conflicts of interest.

## 2. Related Work

### 2.1. Vision-Language-Action Models

The paradigm of robotic learning has shifted from specialized, modular pipelines to unified VLA architectures, which integrate pre-trained vision encoders with language decoders to predict robot actions directly from visual observations and textual instructions. Early works like RT-1 (Brohan et al., 2022) demonstrated the efficacy of tokenizing robotic actions, while RT-2 (Zitkovich et al., 2023) and PaLM-E (Driess et al., 2023) scaled this approach by leveraging internet-scale Vision-Language Models (VLMs) to achieve remarkable semantic generalization and reasoning. More recently, open-source initiatives such as OpenVLA (Kim et al., 2025b) and $\pi_0$ (Black et al., 2024) have adopted more efficient backbones to democratize VLA research. However, standard VLAs inherit a prohibitive computational burden by indiscriminately processing dense token grids where irrelevant background noise equal priority to critical end-effectors. This redundancy not only incurs prohibitive overhead due to the quadratic complexity of the attention mechanism, but also compromises the model's accuracy and generalization capabilities.

### 2.2. Efficient VLA Perception

**Visual Token Pruning.** Recent acceleration strategies seek to minimize the visual token budget through importance-based filtering or temporal reuse mechanisms (Kong et al., 2025). SparseVLM (Zhang et al., 2025) leverages training-free cross-modal attention to prune tokens, yet this semantic-driven selection inherently biases the model toward salient

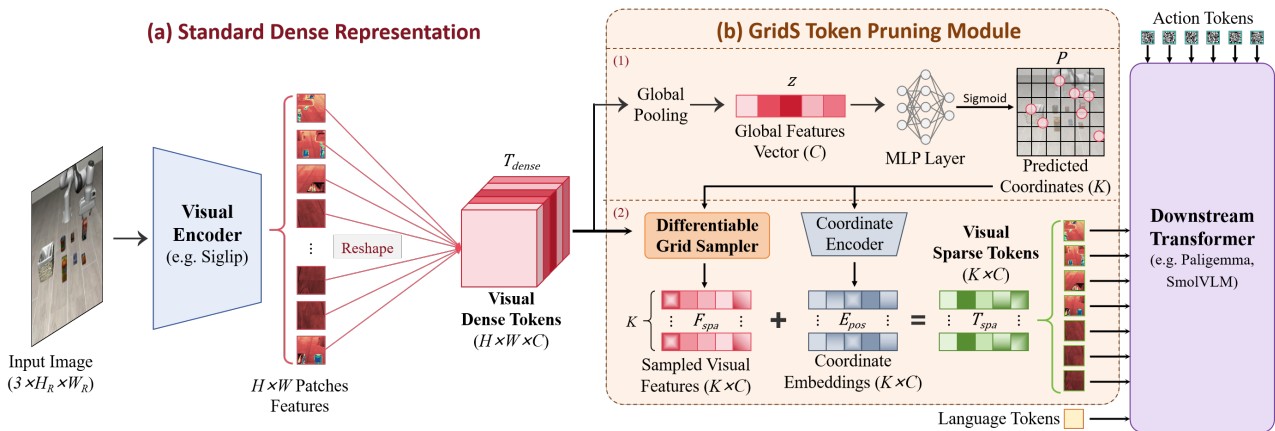

*Figure 3.* Overview of the GridS Token Pruning framework. **(a) Standard Dense Representation**: An input image ($H_R$ and $W_R$ denote the original image resolution) is processed by a visual encoder with ViT embeddings (Dosovitskiy et al., 2021) to generate dense visual tokens ($H \times W \times C$), capturing full spatial details. **(b) GridS Token Pruning Module**: This module identifies salient regions to sample a sparse set of visual tokens ($K \times C$), which includes two Stage: (1) Global Coordinate Prediction, and (2) Grid Sampling with Geometry Injection. By ensuring the token count is significantly smaller than the dense spatial resolution ($K \ll H \times W$), it achieves efficient representation for the downstream Transformer.

objects while neglecting geometrically critical but semantically subtle regions. VLA-Cache (Xu et al., 2025b) exploits temporal redundancy by caching visual features across timesteps, but its reliance on discrete update thresholds over a fixed grid often fails to capture fine-grained geometric deformations during dynamic interaction. EfficientVLA (Yang et al., 2025) employs a learnable policy to dynamically drop background patches, but the resulting ragged tensors severely degrade hardware throughput due to inefficient memory coalescing. To sum up, these approaches are collectively constrained by their reliance on discrete selection over a fixed grid, introducing inevitable quantization errors that sacrifice the sub-patch spatial fidelity essential for precision control.

**Asynchronous Execution.** Alternatively, system-level optimizations leverage asynchronous architectures to decouple the computationally intensive vision encoder from the high-frequency policy head, effectively masking perception latency during real-time deployment (Li et al., 2026b). Representative frameworks such as SmolVLA (Shukor et al., 2025), RTC (Black et al., 2026), and VLASH (Tang et al., 2025), employ multi-threaded pipelines or model compression strategies to parallelize visual processing while maintaining high control frequencies. However, this temporal decoupling inevitably introduces state staleness, where the policy actuates based on outdated observations, resulting in significant performance degradation and instability in highly dynamic manipulation tasks.

### 2.3. Continuous and Deformable Sampling

Traditional Convolution Neural Network and Vision Transformer performs vector embedding on a standard grid (Doso-

vitskiy et al., 2021). To overcome the limitations of fixed grids, deformable mechanisms (Dai et al., 2017; Zhu et al., 2021) and the Vision Transformer with Deformable Attention (DAT) (Xia et al., 2022) introduced learnable offsets, allowing the model to shift attention to flexible spatial locations. In addition, research in implicit neural representations, such as LIIF (Chen et al., 2021) and NeRF (Mildenhall et al., 2021), has demonstrated that modeling images as continuous functions enables feature recovery at arbitrary sub-pixel coordinates.

## 3. GridS Pruning

In this section, we first analyze the computational bottlenecks inherent in standard VLA representations. We then present the **Differentiable Grid Sampler (GridS)**, detailing its pipeline and provide a mathematical derivation of the continuous sampling mechanism. Finally, we outline the end-to-end training objectives.

### 3.1. Efficiency Bottleneck in Standard VLAs

Standard VLA architectures typically employ a frozen Vision Transformer (e.g., SigLIP (Zhai et al., 2023) or DINOv2 (Oquab et al., 2024)) to encode visual observations. As illustrated in Fig. 3 (a), an input image $I \in \mathbb{R}^{3 \times H_R \times W_R}$ is patchified and encoded into a dense feature grid $T_{dense} \in \mathbb{R}^{H \times W \times C}$, where $H \times W$ represents the number of patches (for instance, DINOv2 generate $16 \times 16 = 256$ tokens).

While comprehensive, this dense representation introduces significant redundancy. Since the computational complexity of the downstream Transformer scales quadratically with sequence length ($O(N^2)$), processing hundreds of visual to-

kens, most of which represent non-informative background noise, incurs prohibitive computational costs. This necessitates a pruning mechanism that can effectively discard irrelevant regions while preserving fine-grained details essential for robotic manipulation.

### 3.2. The GridS Pipeline

To address above limitations, we propose GridS, a plug-and-play module designed to compress dense features into a compact set of active tokens. As shown in Fig. 1 (b), the pipeline can divided to two stages: (1) Global Coordinate Prediction, and (2) Grid Sampling with Geometry Injection.

**Global Coordinate Prediction.** Effective active grid sampling requires an understanding of the scene's global semantic structure. Instead of processing local patches in isolation, we first aggregate the entire feature map $T_{dense}$ via global average pooling to obtain a context vector $z \in \mathbb{R}^C$:

$$z = \frac{1}{H \times W} \sum_{i=1}^{H} \sum_{j=1}^{W} T_{dense}^{(i,j)}. \tag{1}$$

This operation provides a summary of the current observation, enabling the model to make informed decisions about where to attend.

Furthermore, we employ a lightweight MLP to predict the locations of the most discriminative regions based on the context $z$ (see Fig. 3 (b-1)). Crucially, rather than outputting discrete grid indices, the network predicts $K$ sets of continuous, normalized coordinates $P \in [0,1]^{K \times 2}$ as $P = \sigma(\text{MLP}(z))$, where $\sigma$ is the Sigmoid function ensuring coordinates remain within the image bounds, and $K \ll (H \times W)$ represents the target number of active tokens.

**Grid Sampling with Geometry Injection.** As shown in Fig. 3 (b-2), we sample from the predicted grid points in the dense tokens (detailed in Sec. 3.3) to generate sparse tokens $F_{spa} \in \mathbb{R}^{K \times C}$, whereas the spatial structure of the original grid is disrupted. To restore geometric awareness, we use a Coordinate Encoder to map the predicted coordinates $P$ into position embeddings $E_{pos} \in \mathbb{R}^{K \times C}$. These embeddings are added to the sampled features to form the final Visual Sparse Tokens ($T_{spa}$), which are then concatenated with language and action tokens for the downstream Transformer.

### 3.3. Differentiable Bilinear Sampling

The core innovation of GridS lies in how it extracts features from the predicted coordinates $P$. Unlike discrete selection strategies that suffer from quantization errors, we employ Differentiable Bilinear Sampling.

As illustrated in Fig. 4, let $P(x,y)$ be a predicted continuous

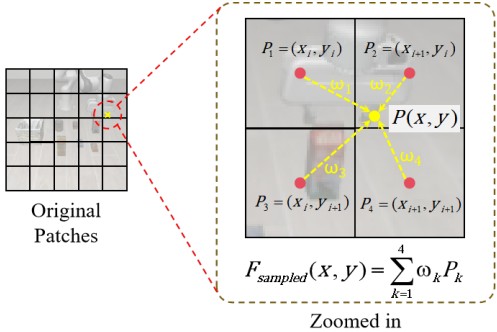

Original Patches — Zoomed in

*Figure 4.* **Differentiable Bilinear Sampling.** To extract features at a continuous coordinate $P(x,y)$, the module computes a weighted interpolation of the four nearest integer neighbors. This operation enables sub-pixel feature extraction and ensures the sampling process is differentiable.

coordinate in the feature map space. The sampling point typically does not align with the integer grid centers. To achieve sub-patch level accuracy, we define the value of the sampled token $F_{sampled}(x,y)$ as a weighted interpolation of its four nearest neighbors on the original grid.

Let the top-left neighbor be $P_1 = (x_i, y_i) = (\lfloor x \rfloor, \lfloor y \rfloor)$. The offsets to the float coordinate are given by $\Delta_x = x - x_i$ and $\Delta_y = y - y_i$. The interpolation weights $\omega_i$ for the four neighbors $\{P_1, P_2, P_3, P_4\}$ are computed as:

$$\begin{aligned} \omega_1 = (1-\Delta_x)(1-\Delta_y), \quad &\omega_2 = \Delta_x(1-\Delta_y), \\ \omega_3 = (1-\Delta_x)\Delta_y, \quad &\omega_4 = \Delta_x\Delta_y. \end{aligned} \tag{2}$$

The final sampled feature is the weighted sum of the neighbor features:

$$F_{sampled}(x,y) = \sum_{k=1}^{4} \omega_k \cdot P_k. \tag{3}$$

This mechanism allows the model to extract sub-patch features. For instance, if a grasp point lies exactly on the edge shared by two patches, GridS can sample the exact boundary by assigning balanced weights to neighbors, whereas discrete selection would be forced to choose just one side.

**Differentiability and Gradient Flow.** A critical property of above bilinear formulation is that it is fully differentiable with respect to the coordinates $(x,y)$, which enables the model to undergo adaptive pruning driven by data. Since the weights $\omega_k$ are linear functions of $x$ and $y$, the gradients can flow back from the task loss $\mathcal{L}$ to the coordinate predictor:

$$\frac{\partial \mathcal{L}}{\partial x} = \sum_{k=1}^{4} \frac{\partial \mathcal{L}}{\partial F_{sampled}} \cdot P_k \cdot \frac{\partial \omega_k}{\partial x}, \tag{4}$$

which implies that the MLP Layer for coordinate generation is not using a heuristic; it is actively trained by the downstream task to move sampling points toward regions that minimize the action prediction error.

*Table 1.* Comparison of Model Efficiency and Performance on LIBERO dataset (Liu et al., 2023). We report efficiency metrics (visual tokens, FLOPs, completion time) and success rate across different task categories. We follow VLASH (Tang et al., 2025) to calculate the average duration and steps of task completion. All efficiency calculations are evaluated on a single RTX Pro6000 GPU and in 32 language tokens. The symbol "$^\dagger$" represents the training-free pruning methods, which forced to operate post-training at test time, leading to inherent performance degradation. (We note that VLA-Cache is not strictly a pruning method, but a token-cached speeding-up method. However, since many pruning methods are also compared with it, we also provide the result here.)

| Model | Efficiency ($\downarrow$) | | | LIBERO Accuracy (%) | | | | | |
| --- | --- | --- | --- | --- | --- | --- | --- | --- | --- |
| | Vis. Tokens | FLOPs (G) | Time (s) | Spatial | Object | Goal | Long | Avg. SR | $\Delta$SR |
| OpenVLA (Kim et al., 2025b) | 256 | 1956.19 | 26.08 | 84.7 | 88.4 | 79.2 | 53.7 | 76.5 | - |
| CogACT (Li et al., 2024) | 256 | 1891.24 | 13.80 | 97.2 | 98.0 | 90.2 | 88.8 | 93.6 | - |
| SmolVLA (Shukor et al., 2025) | 64 | 306.78 | 8.91 | 81.3 | 92.9 | 85.8 | 55.8 | 79.0 | - |
| OpenVLA-OFT (Kim et al., 2025a) | 512 | 3922.56 | 17.43 | 97.6 | 94.2 | 95.2 | 92.0 | 94.8 | - |
| $\pi_0$ (Black et al., 2024) | 256 | 216.01 | 8.17 | 97.2 | 98.8 | 96.0 | 85.6 | 94.4 | - |
| $\pi_0$ + FastV$^\dagger$ (Chen et al., 2024b) | 100 | 143.59 | 7.32 | 97.0 | 98.4 | 93.8 | 82.4 | 92.9 | -1.5 |
| $\pi_0$ + SparseVLM$^\dagger$ (Zhang et al., 2025) | 100 | 150.30 | 7.48 | 93.4 | 98.0 | 91.2 | 76.6 | 89.8 | -4.6 |
| $\pi_0$ + VLA-Cache$^\dagger$ (Xu et al., 2025b) | 256 | 188.12 | 7.52 | 95.2 | 97.6 | 96.6 | 85.6 | 93.8 | -0.6 |
| $\pi_0$ + GridS$_{16}$ | 16 | 51.65 | 6.05 | 98.0 | 99.2 | 96.4 | 90.2 | 96.0 | +1.6 |
| $\pi_0$ + GridS$_4$ | 4 | 43.61 | 5.86 | 96.6 | 99.4 | 96.4 | 89.6 | 95.5 | +1.1 |
| $\pi_{0.5}$ (Intelligence et al., 2025) | 256 | 249.76 | 8.54 | 98.4 | 98.0 | 97.6 | 92.8 | 96.7 | - |
| $\pi_{0.5}$ + GridS$_{16}$ | 16 | 83.84 | 6.76 | 98.6 | 98.8 | 98.4 | 95.2 | 97.7 | +1.0 |
| $\pi_{0.5}$ + GridS$_4$ | 4 | 75.73 | 6.53 | 97.4 | 99.0 | 97.4 | 92.8 | 96.7 | 0.0 |

## 3.4. Training Objectives and Joint Optimization

Existing token pruning methods are typically applied post-training, which is described as a training-free method. Rather than an inherent advantage, this "training-free" property is a forced compromise due to their non-differentiable designs, which inevitably degrades downstream performance. GridS integrated directly between the vision encoder and transformer, it is jointly optimized with the VLA policy during standard fine-tuning. It trains end-to-end using only the primary task loss—without any auxiliary supervision or ground-truth attention maps—making it seamlessly compatible with diverse paradigms, including auto-regressive and flow-matching models. Since downstream fine-tuning is already a prerequisite for VLAs, GridS introduces no extra pipeline steps. By heavily compressing visual tokens early in the forward pass, it significantly reduces the computational footprint and accelerates training compared to standard implementations.

## 4. Experiments

We design experiments to investigate the following questions of GridS:

1. **Comparative Advantage.** Does GridS offer a competitive edge over SOTA autoregressive and flow-matching pruning methods (Sec. 4.1 and Sec. 4.2)?

2. **Generalization Capability.** Does the introduction of GridS perturb the generalization boundaries of the pre-trained model (Sec. 4.2)?

3. **Efficiency.** Does GridS enhance efficiency and mitigate computational overhead (Sec. 4.3)?

4. **Mechanism of High Compression.** What enables GridS to maintain model capability despite a massive reduction ($> 90\%$) in visual information (Sec. 4.4)?

### 4.1. Simulation Experiments

#### 4.1.1. LIBERO

**Experiments setup.** We conduct experiments on the Libero simulation benchmark (Liu et al., 2023), specifically utilizing its four distinct suites ( *Spatial*, *Object*, *Goal*, and *Long*) that contain 10 tasks each. These suites are designed to evaluate specific robotic capabilities: spatial understanding, object recognition, goal-directed behavior, and long-horizon planning, respectively. We select $\pi_0$ (Black et al., 2024) and $\pi_{0.5}$ (Intelligence et al., 2025), two flow-matching method as our baseline models. Leveraging GridS, we aggressively prune the visual tokens from the baseline count of 256 down to set budgets of 16, and 4. We evaluate the performance across all four suites, reporting the accuracy for the tasks, as well as the computational cost (FLOPs) and inference time associated with each token density.

**Main Results.** Evaluations on LIBERO (Tab. 1) reveal a critical flaw in traditional post-training pruning: discretely dropping patches (FastV, SparseVLM) destroys spatial continuity, degrading $\pi_0$ success rates by $1.5\%$ and $4.6\%$. In contrast, GridS resolves this efficiency-performance trade-off through joint optimization. Compressing the standard 256 tokens down to 16 reduces FLOPs by $\sim 76\%$ (to 51.6 G) and execution time by $> 2.1$s, while actually yielding a

*Table 2.* Evaluation on ALOHA benchmark ([Zhao et al., 2023](#)). We report the average success rates and enviroment rewards across 3 random seeds. The performance is evaluated on a RTX Pro6000 GPU.

| Method | Visual Tokens | Times (s) | Env.↑ Rewards | Avg.↑ Accuracy (%) | Transfer Cube | | Insertion | |
|---|---|---|---|---|---|---|---|---|
| | | | | | Scripted | Human | Scripted | Human |
| $\pi_0$ ([Black et al., 2024](#)) | 256 | 7.04 | **2.44** | 86.3 | 100.0 | 96.9 | 91.4 | 56.7 |
| $\pi_0$ + GridS | **16** | **6.32** | 2.38 | **87.0** | 100.0 | 96.9 | 86.9 | 64.2 |

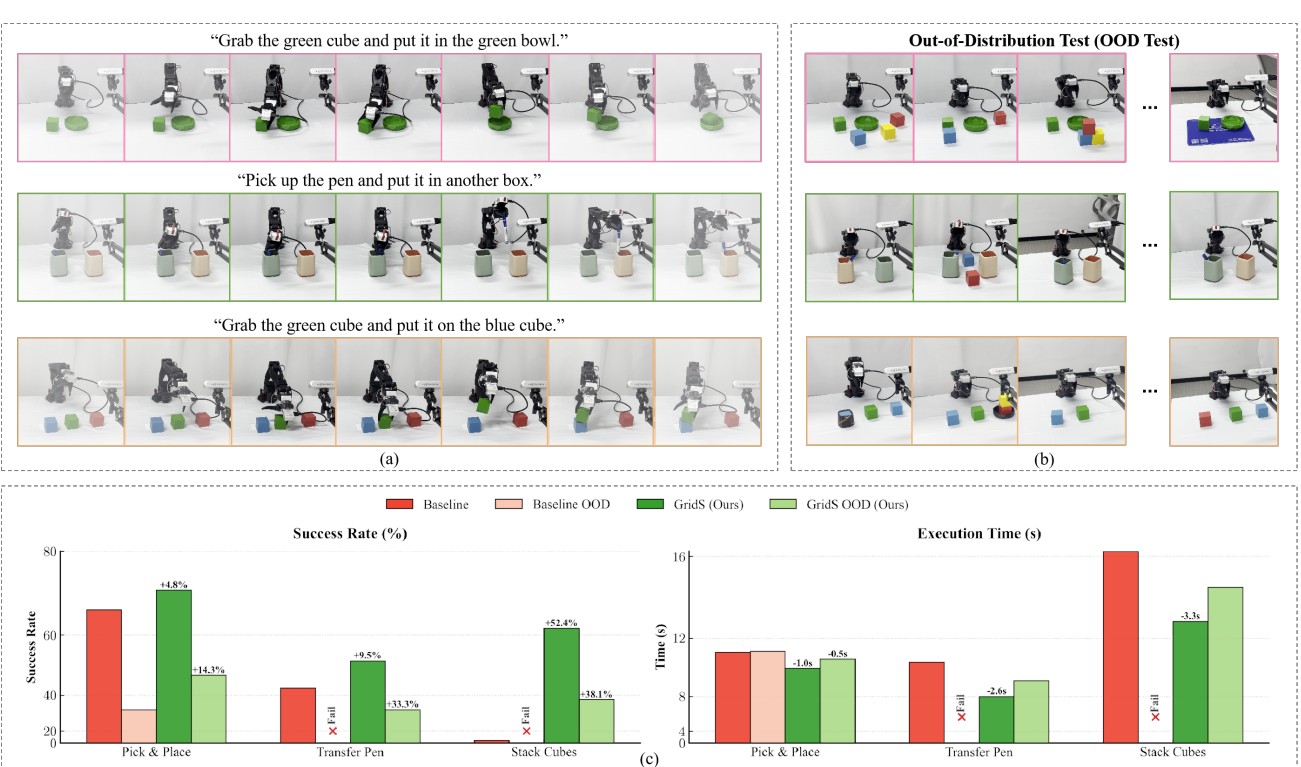

*Figure 5.* **Real-world evaluation on the SO100 robot arm.** (a) Execution rollouts of three language-conditioned tasks: *Pick & Place*, *Stack Cubes*, and *Transfer Pen*. (b) The corresponding Out-of-Distribution (OOD) test scenarios, featuring unseen distractor objects and variable spatial arrangements. We schemed 21 different OOD scenarios. (c) Quantitative comparison of Success Rate (%) and Execution Time (s). Our proposed method, **GridS**, consistently outperforms the baseline across all tasks. Notably, in the challenging *Stack Cubes* task, GridS achieves a **+52.4%** increase in success rate and demonstrates superior generalization in OOD settings where the baseline frequently fails.

+1.6% absolute improvement in average success rate. Notably, this gain peaks at +4.6% in the Long-horizon suite. Since long-horizon tasks are highly vulnerable to error accumulation, this confirms that our continuous resampling mechanism effectively filters visual distractors to produce robust state representations. Furthermore, these empirical advantages scale consistently to the stronger $\pi_{0.5}$ backbone.

### 4.1.2. ALOHA

**Experiments setup.** Built upon MuJoCo ([Todorov et al., 2012](#)), the ALOHA benchmark ([Zhao et al., 2023](#)) targets fine-grained bimanual manipulation through two primary tasks: *Transfer Cube* and *Bimanual Insertion*. The dataset comprises 50 demonstrations per task, collected via scripted policies or human teleoperation, with an episode horizon

of 400 steps (8 seconds). We evaluate each method using three distinct random seeds. For both the baseline and GridS-integrated models, we conducted a total of at least 300 evaluation episodes across the two tasks (covering 100 unique initializations) and reported the average success rate.

**Main Results.** As shown in Tab. [2](#), GridS demonstrates exceptional efficiency-performance trade-offs. With only 16 visual tokens (vs. 256 in baseline), our method reduces inference latency by $\sim 10\%$ while achieving a better average success rate of 87.0%. Specifically, the success rate of GridS on the Transfer Cube task remains unchanged, which can be attributed to overfitting. In the challenging Insertion (Human) task, GridS significantly outperforms the baseline +7.5%, indicating improved robustness to the variability

inherent in human demonstrations. This phenomenon becomes even more pronounced in real-world settings, a topic that will be elaborated upon in the following subsection.

## 4.2. Real-world Experiments

**Experimental Setup and Dataset.** We conduct our experiments on a standard Linux workstation equipped with an NVIDIA RTX 3090 (24GB) GPU. The policy is based on the SmolVLA architecture (Shukor et al., 2025) and is trained with a batch size of 16 for 50,000 optimization steps. We evaluate the policy on three real-world manipulation tasks executed on a SO100 robot (Cadene et al., 2024), utilizing a dataset collected to cover varying complexities. The dataset is constructed as follows: (1) *Pick & Place*: 83 episodes where the robot grasps a green cube from two distinct initial positions and places it into a green bowl; (2) *Transfer Pen*: 74 episodes involving picking up a pen and placing it into a container; (3) *Stack Cubes*: 75 episodes of stacking a green cube onto a blue cube, comprising 50 trajectories targeting the blue cube on the right side of the SO100 and 25 targeting the left side. While Fig. 5 (a) illustrates the ideal environment used for data collection and routine testing, we rigorously evaluate generalization using a benchmark of 21 distinct OOD scenarios (see Fig. 5 (b)), characterized by severe visual distractors and novel object arrangements to stress-test the policy's robustness.

**Main Results.** As reported in Fig. 5 (c), GridS consistently outperforms the baseline across all tasks. This performance gap is particularly pronounced in the most challenging Stack Cubes task, where our method achieves a 60.0% success rate compared to the baseline's ∼7.6%, yielding a substantial +52.4% improvement. Crucially, GridS exhibits superior robustness in OOD scenarios; while the baseline completely fails in the OOD settings for Transfer Pen and Stack Cubes, GridS maintains capable performance. Furthermore, our sparse tokenization translates to tangible efficiency gains, reducing the average execution time by up to 3.3s (Fig. 5 (c), right). These results confirm that GridS effectively eliminates redundancy to enhance both policy robustness and inference speed.

## 4.3. Ablation Study

### 4.3.1. TOKEN COUNT

**LIBERO.** We analyze the impact of the sampled token number $K$ on model efficiency and success rates (Tab. 1). While GridS generally improves performance, we observe task-specific sensitivities at extreme sparsity ($K = 4$). First, Spatial shows a 1.2% decline on $\pi_{0.5}$. Although GridS can identify the most suitable location points based on image features, it still suffers from geometric ambiguity; precise spatial relationships require fine-grained boundary infor-

*Table 3.* Ablation study of success rate of stacking cubes in real-world with different tokens. Results with SO100 robot.

| Method | Vis. Tokens | Success Rate (%) | OOD SR (%) |
|---|---|---|---|
| GridS | 4 | 0.0 | 0.0 |
| | 8 | 28.5 | 19.1 |
| | **16** | **61.9** | **38.1** |
| | 32 | 19.0 | 0.0 |
| Baseline | 64 | 9.5 | 0.0 |

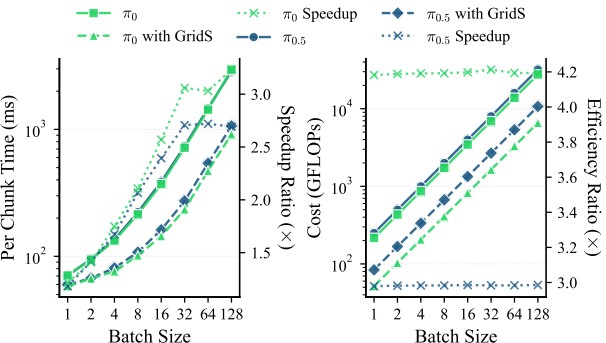

*Figure 6.* **Performance Analysis.** We compare the inference latency (left) and computational cost (right) of the baseline method versus our proposed GridS pruning (16 tokens) across varying batch sizes. Solid and dashed lines denote the absolute values (left y-axis), while the dotted lines indicate the relative speedup and efficiency ratios (right y-axis).

mation that may be lost when representing an object with only few tokens. Second, Long performance drops in the sparsest setting ($K = 4$). Since long-horizon tasks are vulnerable to error accumulation, transient sampling misses in intermediate frames—more likely under strict token budgets—can disrupt sequential execution. However, Object remains robust (peaking at 99.4% with $K = 4$), validating that core semantic recognition requires minimal visual data. This can be attributed to the high instruction dependency of the task, where the sequence and position of object placement are relatively static. As a result, the model is more inclined to memorize these tokens during fine-tuning instead of comprehending the underlying logic.

Given the severe overfitting tendencies associated with the LIBERO dataset, deriving fine-grained insights solely from these experiments is challenging. Consequently, we conducted further ablation studies in real-world settings to validate our method. We have reported more ablation experiments in the appendix.

**Real-world.** To investigate the impact of token density on fine-grained manipulation, we evaluate the Stacking Cubes task across varying token budgets $K \in \{4, 8, 16, 32\}$.

As shown in Tab. 3, the results exhibit a distinct inverted U-shaped trend, pinpointing $K = 16$ as the optimal trade-

*Table 4.* Average training time per step (batch size equals 128).

| Model | Time (s/step) $\downarrow$ | Speedup $\uparrow$ |
|---|---|---|
| $\pi_0$ (Baseline) | 14.32 | 1.0× |
| $\pi_0$ + **GridS** | **4.21** | **3.4×** |
| $\pi_{0.5}$ (Baseline) | 12.83 | 1.0× |
| $\pi_{0.5}$ + **GridS** | **4.41** | **2.9×** |

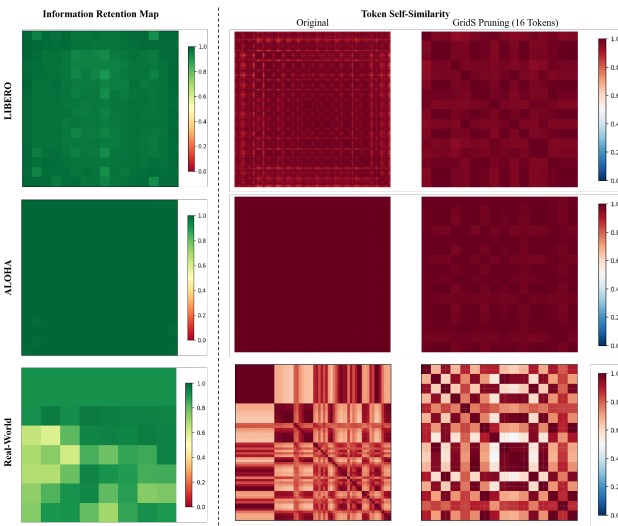

*Figure 7.* Visualization of Information Retention and Sampling Efficiency. We evaluate GridS on LIBERO, ALOHA, and Real-World data. Left: Information Retention Maps demonstrate that our sampling strategy maintains high information retention (green), effectively covering the original feature space. Right: Token Self-Similarity matrices reveal that while original features suffer from high spatial redundancy.

off between semantic sufficiency and redundancy reduction. At extreme sparsity ($K = 4$), the policy fails completely, indicating an information bottleneck where the representation lacks the geometric resolution required for precise alignment. Conversely, increasing $K$ beyond the optimal point leads to a sharp performance degradation (19.0% at $K = 32$), trending towards the poor performance of baseline. This counter-intuitive drop suggests that excessive tokens introduce irrelevant background noise, which overwhelms the policy in precision-sensitive scenarios. By employing bilinear interpolation to compress the token count by a factor of 4, GridS ($K = 16$) functions as an effective feature sampler. This downsampling process significantly filters out redundancy and maximizes the signal-to-noise ratio, thereby facilitating robust manipulation.

We further evaluated robustness across 21 diverse real-world OOD scenarios, where $GridS_{16}$ demonstrates superior generalization (38.1%) compared to the complete failure of the baseline and other settings. Reducing the budget to $K = 8$ causes the policy to suffer from severe positional bias, where successful manipulation is restricted exclusively to objects initialized on the left side due to insufficient spatial coverage. Conversely, increasing to $K = 32$ introduces excessive visual noise that impairs terminal state recognition; specifically, while it can stack the cubes, the distraction prevents it from deciding to release the gripper, leading to task failure.

### 4.3.2. COMPUTATIONAL EFFICIENCY ANALYSIS

We benchmark the per-chunk inference time and computational cost across varying batch sizes from 1 to 128 on both $\pi_0$ and $\pi_{0.5}$ backbones (Fig. 6). Theoretically, GridS reduces the compute load, achieving a consistent FLOPs reduction ratio of approximate 4.2× for $\pi_0$ and 3.0× for $\pi_{0.5}$. Practically, we note that the wall-clock speedup is less pronounced at test-time inference (one batch), showing a modest ∼1.2× gain. This is because the dense baseline is already highly optimized by JAX (Bradbury et al., 2018) compilation, leaving the runtime dominated by fixed kernel overheads rather than computation. Consequently, the benefits of GridS are most substantial in high-throughput settings where GPU compute is saturated, scaling rapidly to a 3.2× speedup at 128 batch size. The benefits of our method are further magnified throughout the training process. As Tab. 4, our method significantly accelerates the

optimization process, achieving a substantial 2.9× and 3.4× speedup for $\pi_{0.5}$ and $\pi_0$ respectively at a batch size of 128.

### 4.4. Why GridS Works?

GridS adaptively samples global contextual cues using a learnable MLP and bilinear interpolation. Fig. 7 (right) illustrates the resulting reduction in spatial redundancy: while original feature maps display dense off-diagonal correlations, GridS yields a clean, diagonal-dominant structure, confirming that the bilinear sampling transforms redundant inputs into a compact, orthogonal token set.

We visualize the corresponding semantic coverage via Information Retention maps (Fig. 7 left). A more detailed calculation method for retention map can be found in the appendix I. In visually simple environments like standard ALOHA, the model easily covers the clean scene, achieving near-uniform retention (0.9994). However, retention scores naturally drop in LIBERO (0.9709) and noisy Real-World scenarios (0.8610). While a strictly uniform map represents the theoretical ideal of zero information loss, preserving all visual input actually degrades performance in cluttered domains. To achieve high success rates in complex environments, the policy must discard irrelevant distractors. Therefore, the low-retention areas are not a flaw; they correctly reflect GridS actively filtering out background noise to concentrate its capacity on critical dynamic geometries. This inherent spatial regularization explains why GridS yields massive out-of-distribution (OOD) improvements over full-

token baselines. We have analyzed more possible reasons in the Appendix G and K.

## 5. Conclusion

In this work, we presented the Differentiable Grid Sampler (GridS), a plug-and-play module that enhances VLA efficiency by reformulating token compression as continuous, task-aware resampling. Crucially, our extensive experiments validate that complex manipulation tasks can be successfully executed with as few as 16 visual tokens, establishing a new benchmark for the lowest feasible token count reported to date. By replacing fixed-grid redundancy with high-fidelity sparse sampling, GridS not only reduces computational costs by 76% but also achieves significant gains in out-of-distribution robustness (+28.6%).

## Acknowledgement

This work was supported in part by the Australian Research Council under Projects DP240101848 and FT230100549. In addition, we express our gratitude to Jackie L. for his helpful discussions on this paper and to Iroha S. for her spiritual support.

## Impact Statement

The primary goal of this work is to enhance the computational efficiency of Vision-Language-Action models. By significantly reducing inference costs, our approach contributes to 'Green AI' initiatives by lowering energy consumption and democratizes access to advanced robotic capabilities on consumer-grade hardware. Beyond general considerations regarding robotic automation, we do not foresee any specific negative consequences stemming directly from this work.

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

# Appendix

## A. Real-World Experimental Setup

We evaluate our method on a real-world teleoperation platform designed for precise manipulation tasks. Figure 8 illustrates the hardware configuration used for policy deployment.

**Robot Manipulators.**    We utilize the **LeRobot SO-100**, a 6-DoF low-cost open-source robotic arm, as the follower robot for execution. For data collection, we employ a leader-follower teleoperation scheme where a second identical SO-100 arm captures precise joint positions. These positions are mapped directly to the follower arm to ensure smooth demonstration recording.

**Vision System.**    Our vision-language-action model processes visual feedback from a multi-view camera setup. **It is important to note that while the hardware supports depth sensing, our policy operates exclusively on RGB images.**

- **Front(Third-Person) View:** An **Intel RealSense D435** camera is mounted diagonally above the robot at approximately 45 degrees relative to the workspace. This viewpoint provides a holistic understanding of the scene layout and object positions relative to the robot base.

- **Wrist(Egocentric) View:** An **Intel RealSense D405** camera is mounted directly on the robot's wrist. This enables close-up observation of the end-effector gripper and the final contact between the gripper and objects.

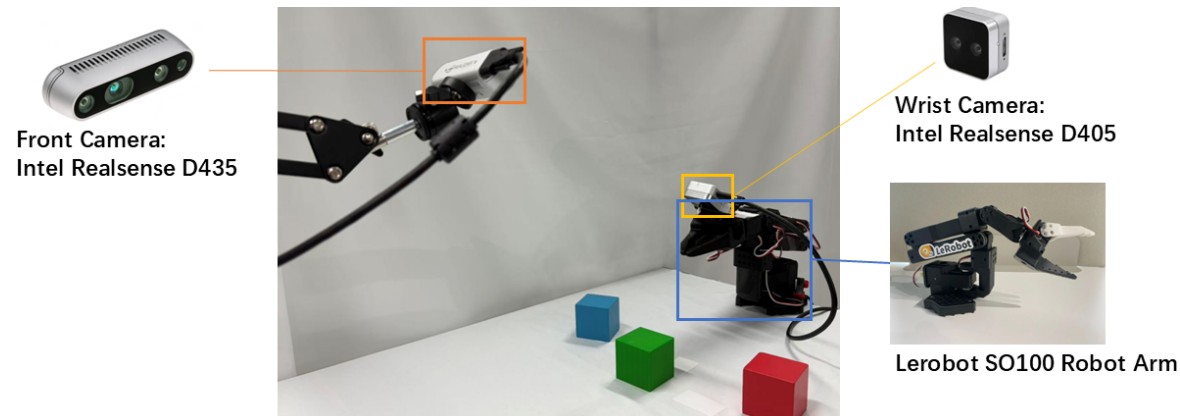

*Figure 8.* **Real-World Hardware Setup.** The image displays the LeRobot SO-100 follower arm used for policy execution. Visual inputs come from a fixed Intel RealSense D435 providing global scene context and a wrist-mounted Intel RealSense D405 capturing fine-grained local details. The policy operates using only RGB streams from these sensors.

**Real World Tasks.**    We provide comprehensive video demonstrations comparing our proposed GridS against the unpruned dense baseline SmolVLA and a standard token pruning baseline. All real-world experiments utilize the LeRobot SO-100 setup described above.

- **Pick & Place (83 episodes):** The robot grasps a green cube from two distinct initial positions and places it into a green bowl. This task serves as a benchmark for motion quality. Thanks to the differentiable nature of GridS, the policy produces smooth trajectories, effectively filtering out high-frequency control noise while maintaining a stable path towards the target.

- **Transfer Pen (74 episodes):** A high-precision task involving picking up a thin marker pen and placing it into a cup container. This task rigorously tests the model's fine-grained accuracy and capability for delicate manipulation. Instead of mechanically memorizing training trajectories, GridS demonstrates robust generalization by precisely sampling features from the small target object (the pen). This allows for accurate end-effector alignment even with slight positional variations, ensuring the gripper successfully interacts with the thin geometry.

- **Stack Cubes (75 episodes):** A **multi-stage task** requiring the robot to stack a green cube onto a blue cube. The dataset includes 50 trajectories targeting the right side and 25 targeting the left side. This task demands strong **spatial reasoning** and the ability to handle **longer-horizon sequences** (approach → grasp → align → stack). The model must understand the relative 3D relationship between the held object and the target base, maintaining this spatial context over a longer execution timeline than simple pick-and-place tasks.

**Out-of-Distribution (OOD) Evaluation.** To rigorously evaluate model robustness beyond the training distribution, we established over 20 distinct out-of-distribution (OOD) tests for each task. Although training data was collected under consistent conditions, these evaluation scenarios introduce severe visual and spatial perturbations. We categorized these perturbations into seven key example types to stress-test model generalization capabilities (as shown in Figure 9):

- **Cluttered Background:** Replace clean tabletops with patterned fabrics and remove backdrops from front-view backgrounds to test visual attention.

- **Novel Target Instances:** Place standard training cubes on unfamiliar backgrounds to introduce confusion.

- **Removing Training Priors:** Eliminate auxiliary objects (e.g., standard background cubes) present throughout training to ensure the model relies on targets rather than environmental landmarks.

- **Unknown Distractions:** Introduce diverse, irrelevant clutter (e.g., tools, snacks) into the workspace to create visual interference.

- **Spatial extrapolation:** Position target objects in locations not covered by demonstration data.

- **Adversarial layouts:** Place distractors near target objects or along optimal trajectory paths.

- **Compound perturbations:** Combine visual clutter, novel objects, and unseen spatial layouts simultaneously to maximize task difficulty.

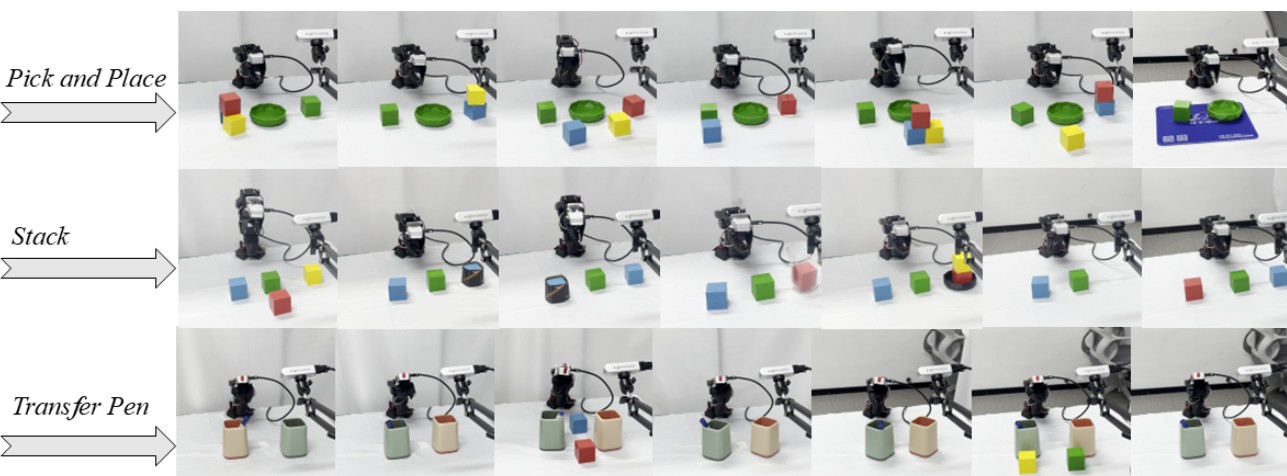

*Figure 9.* **Visualization of OOD Scenarios.** We selected seven examples per task to demonstrate how we evaluated the strategy across over 20 scenarios categorized into seven types of perturbations: cluttered backgrounds, novel objects, removed training scenes, and unseen spatial layouts. Across these settings, GridS demonstrated stronger robustness compared to baseline models.

## B. Hyperparameter Settings

We present the detailed training hyperparameters used for fine-tuning the VLA models in our experiments in Table 5. To ensure rigorous and fair comparisons, we employ a unified set of hyperparameters across all methods—including the unpruned baseline model, the standard token pruning approach, and our proposed GridS model—which are also applied to real-world manipulation tasks. These parameters are empirically optimized to balance training stability and convergence efficiency while effectively preventing overfitting.

*Table 5.* Hyperparameters derived from configuration.

| Model | SmolVLA | $\pi_0$ (LIBERO/ALOHA) | $\pi_{0.5}$ (LIBERO) |
|---|---|---|---|
| *Training Configuration* | | | |
|    Batch Size | 16 | 32 | 128 |
|    Training Steps | 50,000 | 30,000 | 30,000 |
|    Action Chunk Size | 50 | 10 | 10 |
|    Input Image Resolution | $512 \times 512$ | $224 \times 224$ | $224 \times 224$ |
| *Optimizer (AdamW)* | | | |
|    Learning Rate (LR) | 1e-4 | 5e-5 | 5e-5 |
|    Betas | | [0.9, 0.95] | |
|    Weight Decay | | 1e-10 | |
|    Grad Clip Norm | 10 | 1.0 | 1.0 |
| *Learning Rate Scheduler* | | | |
|    Type | | Cosine Decay with Warmup | |
|    Warmup Steps | 1,000 | 10,000/1,000 | 10,000 |
|    Peak Learning Rate | 1e-4 | 5e-5 | 5e-5 |
|    Decay Learning Rate | 2.5e-6 | 5e-5/2.5e-6 | 5e-5 |
|    Decay Steps | 50,000 | 30,000 / 15,000 | 30,000 |

## C. Video Demonstrations and Qualitative Analysis

We provide comprehensive video demonstrations in the supplementary material to qualitatively validate the results reported in the main paper. These videos offer a side-by-side comparison between our proposed **GridS** and the strongest baseline (SmolVLA) across both standard and Out-of-Distribution (OOD) settings.

**Standard Benchmarks: Efficiency and Smoothness.** In the standard *Pick & Place*, *Transfer Pen*, and *Stack Cubes* tasks, the demonstrations highlight the significant performance gap between the two models.

- **Completion Speed:** Viewers can observe that the GridS-controlled robot completes tasks noticeably faster than the baseline. The reduced computational overhead results in more fluid and continuous motion, avoiding the "stop-and-go" pauses often observed in the baseline execution.

- **Motion Quality:** Despite pruning 90% of the visual tokens, GridS maintains precise end-effector control, effectively eliminating the jitter caused by discrete quantization errors in fixed-grid baselines.

**OOD Scenarios: True Attention vs. Rote Memorization.** The videos also showcase the deployment of both models in the challenging OOD scenarios described in Appendix. The visual comparison reveals a fundamental difference in policy behavior.

- **Genuine Attention vs. Mechanical Memorization:** A critical observation is the difference in failure modes. The baseline often behaves as if executing a **mechanically memorized action sequence**, failing blindly when the target object is shifted or the environment changes. In sharp contrast, GridS demonstrates **genuine visual attention**. It can be seen actively "looking" for the target, dynamically adjusting its trajectory to lock onto the object's actual position rather than replaying a fixed path.

- **Robustness in Complex Scenes:** Under high visual load (e.g., cluttered desks), GridS successfully filters out background noise to focus on the target, maintaining **real-time responsiveness** (low latency). The baseline, conversely, often struggles to distinguish the target from the distractors, leading to freezing or incorrect actuation.

The full video demonstrations can be found in the Github link.

## D. Robustness to Camera Viewpoint Loss

To evaluate the generalization capability and robustness of GridS under sensor failure scenarios, we conducted a "Viewpoint Loss" experiment on the LIBERO benchmark. In this setting, models were trained using full visual observations (Third-

*Table 6.* **Robustness to Viewpoint Loss (Wrist-Only Inference).** Models are trained with both Base and Wrist cameras but evaluated using *only the Wrist camera*. We compare two missing-modality strategies: Masking tokens vs. Black Image input. GridS ($K = 16$) demonstrates superior robustness compared to the Baseline.

| Missing Strategy | Task Suite | Baseline (Acc) | GridS (Acc) |
|---|---|---|---|
| **Token Masking** | Spatial | 0.66 | **0.92** (+0.26) |
| | Object | 0.88 | **0.92** (+0.04) |
| | Goal | 0.76 | **0.94** (+0.18) |
| | Long | 0.30 | **0.70** (+0.40) |
| **Black Image** | Spatial | 0.62 | **0.82** (+0.20) |
| | Object | 0.84 | **0.92** (+0.08) |
| | Goal | 0.62 | **0.68** (+0.06) |
| | Long | 0.18 | **0.44** (+0.26) |

person Front + Egocentric Wrist cameras) but evaluated using only the Wrist camera during inference. We tested two strategies for simulating the missing base camera: (1) Masking: explicitly masking the visual tokens corresponding to the base view; and (2) Black Image: replacing the base view input with a blank black image.

As shown in Table 7, the standard Baseline suffers catastrophic performance degradation when the global view is removed, particularly in spatial reasoning (spatial) and long-horizon (10) tasks (e.g., dropping to 18% success rate in LIBERO-10 with black images). This suggests the Baseline over-relies on the static global view and fails to leverage fine-grained cues from the wrist camera.

In contrast, GridS maintains remarkably high success rates across all tasks despite the missing modality. In the "Masking" setting, GridS achieves 0.92 on Spatial tasks (vs. Baseline 0.66) and 0.70 on Long-horizon tasks (vs. Baseline 0.30). This demonstrates that GridS's active sampling mechanism effectively reduces dependency on specific viewpoints by learning to extract task-relevant features (e.g., object geometry and interaction points) directly from the available wrist-view tokens, ensuring robust control even in strictly egocentric environments.

We further conducted ablation experiments on the wrist-only (black image form) on GridS:

*Table 7.* Success rates on LIBERO dataset using strictly wrist-camera visual inputs (third-person view masked). Performance exhibits high sensitivity and non-linear fluctuations with respect to the token budget $K$.

| Token Budget ($K$) | Spatial | Object | Goal | Long |
|---|---|---|---|---|
| $K = 1$ | 15.2 | 1.8 | 9.0 | 0.8 |
| $K = 4$ | 79.2 | 90.2 | 63.0 | 53.2 |
| $K = 6$ | 59.0 | 67.4 | 57.2 | 44.6 |
| $K = 8$ | 68.0 | 63.8 | 48.8 | 18.6 |
| $K = 10$ | 59.2 | 25.0 | 26.8 | 18.4 |
| $K = 12$ | 76.8 | 83.6 | 56.4 | 42.0 |
| $K = 16$ | **80.6** | 84.8 | 68.0 | 50.8 |
| $K = 32$ | 71.6 | **95.0** | **72.2** | **55.4** |

**Empirical Observations:** The results demonstrate a highly non-linear, oscillating trend rather than a smooth scaling curve. At extreme sparsity ($K = 1$), the policy predictably fails due to severe information loss. Surprisingly, performance peaks dramatically at $K = 4$ across all suites. However, as the token budget slightly increases to intermediate values ($K \in \{6, 8, 10\}$), the success rate experiences a catastrophic drop. The performance then robustly recovers at larger token budgets ($K \geq 12$).

**Possible Interpretations:** While the exact underlying mechanics of this fluctuation remain an open question for future investigation, we hypothesize that it stems from the complex interaction between the constrained field of view (FOV) of the wrist camera and the spatial optimization dynamics of GridS:

- *Geometric Symmetry and Optimization Landscape:* The high-performing budgets (e.g., $K \in \{4, 16, 32\}$) correspond

to numbers that easily form symmetric spatial arrangements (such as $2 \times 2$ or $4 \times 4$ conceptual grids). The coordinate predictor might inherently favor optimizing symmetric, evenly distributed continuous points. Non-standard budgets ($K \in \{6, 8, 10\}$) may force the network into asymmetric local minima during end-to-end training, thereby disrupting spatial awareness.

- *Wrist Camera FOV Bottleneck:* Without the global context from the third-person camera, the wrist camera view is heavily dominated by the gripper and the immediate target object. A minimal budget like $K = 4$ might perfectly align with the core local geometries (e.g., the two gripper tips and two target keypoints). When given slightly more tokens ($K \in \{6, 8, 10\}$), the model may attempt to track out-of-focus background details to infer global goals, leading to attention distraction. When $K \geq 16$, the token capacity becomes sufficient to redundantly encode both local structures and background cues without severe feature conflicts.

## E. More experiments

### E.1. LIBERO-Plus

E.1.1. OVERVIEW OF THE LIBERO-PLUS BENCHMARK

Current VLA models exhibit highly saturated performance on standard robotic manipulation benchmarks such as the original LIBERO. However, these aggregate success rates often mask the inherent vulnerability of the models when deployed in dynamic, real-world environments. To rigorously evaluate the robustness and generalization capabilities of these policies, the LIBERO-PLUS benchmark systematically extends the original dataset by introducing controlled, cross-dimensional perturbations designed to stress-test models against realistic distribution shifts (Fei et al., 2025).

Specifically, LIBERO-PLUS incorporates seven core perturbation dimensions, which are further subdivided into 21 low-level components, to comprehensively assess the robustness of multimodal inputs across vision, state, and language. These dimensions encompass variations in camera viewpoints (e.g., modifying distance, spherical position, and orientation), robot initial states via random joint angle perturbations, and language instructions, which are rewritten using Large Language Models to introduce distracting contexts or alter reasoning complexity. Furthermore, the benchmark rigorously evaluates visual robustness by modifying light conditions (such as diffuse color, direction, and shadows), background textures of scenes and surfaces, sensor noise (simulating motion blur, Gaussian blur, and fog), and object layouts by introducing unseen confounding objects and perturbing target object poses.

Beyond its extensive coverage of perturbation factors, the benchmark is notably large in scale, comprising 10,030 evaluation tasks. To provide fine-grained insights into when and how VLA models fail, LIBERO-PLUS introduces a dynamic difficulty stratification mechanism. All tasks are pre-evaluated using four representative state-of-the-art models and are categorized into five difficulty levels (Level-1 to Level-5) based on the number of models that successfully complete them. In addition to this comprehensive evaluation suite, the benchmark provides a generalized training dataset containing over 20,000 successful trajectories, enabling researchers to investigate whether mixed fine-tuning on generalized data can substantially mitigate performance degradation and enhance policy robustness.

E.1.2. IN-DOMAIN EVALUATION ON THE LIBERO-PLUS BENCHMARK

We conduct an in-domain evaluation where both the dense baseline ($\pi_{0.5}$) and our compressed model ($\pi_{0.5} + \text{GridS}_{32}$) are directly co-trained on the augmented LIBERO-PLUS dataset and tested on its corresponding evaluation suite. The quantitative results across the four suites, broken down by perturbation dimension and difficulty level, are presented in Table 8a.

**1. Remarkable Parameter and Data Efficiency under In-Domain Augmented Training.** When co-trained on the augmented LIBERO-PLUS dataset, the dense baseline ($\pi_{0.5}$) recovers from its previously catastrophic OOD failures (e.g., Camera Viewpoints in Spatial jumps from 67.0% to 94.9%). Given this highly saturated in-domain setting, it is remarkable that GridS successfully retains competitive overall performance across all suites (only a marginal degradation of 1.1% to 3.0% on average). Operating on just 32 visual tokens—an 87.5% reduction in visual density compared to the baseline's 256 tokens—GridS proves that even when the training distribution is highly complex and perturbed, a vast majority of standard visual tokens are mathematically redundant for acquiring dexterous manipulation policies.

**2. The Persistent Advantage of Information Bottleneck in Visual Perturbations.** The most profound observation from this in-domain evaluation (Panel A) is that despite the baseline being explicitly trained on all perturbation types, GridS *still*

*Table 8.* In-Domain Evaluation on the LIBERO-PLUS Benchmark. Both the dense baseline ($\pi_{0.5}$) and the compressed model ($\pi_{0.5} +$ GridS$_{32}$) are co-trained on the augmented dataset. Performance is broken down by perturbation dimension (Panel A) and task difficulty level (Panel B) across four suites.

*(a) LIBERO-Spatial*

| Metric Category | $\pi_{0.5}$ Baseline | $\pi_{0.5}$ + GridS$_{32}$ | $\Delta$ |
|---|---|---|---|
| *Panel A: By Perturbation Dimension* | | | |
| Background Textures | 98.4 | 98.4 | **0.0** |
| Camera Viewpoints | 94.9 | 97.9 | **+3.0** |
| Language Instructions | 82.8 | 77.9 | -4.9 |
| Light Conditions | 92.5 | 96.2 | **+3.7** |
| Objects Layout | 92.2 | 90.1 | -2.1 |
| Robot Initial States | 77.1 | 69.1 | -8.0 |
| Sensor Noise | 94.6 | 96.3 | **+1.7** |
| *Panel B: By Task Difficulty Level* | | | |
| Level 1 (Easiest) | 91.2 | 92.3 | +1.1 |
| Level 2 | 92.5 | 93.4 | +0.9 |
| Level 3 | 91.3 | 90.3 | -1.0 |
| Level 4 | 89.1 | 85.9 | -3.2 |
| Level 5 (Hardest) | 75.4 | 66.0 | -9.4 |
| **Overall** | **90.0** | **88.8** | **-1.2** |

*(b) LIBERO-Object*

| Metric Category | $\pi_{0.5}$ Baseline | $\pi_{0.5}$ + GridS$_{32}$ | $\Delta$ |
|---|---|---|---|
| *Panel A: By Perturbation Dimension* | | | |
| Background Textures | 98.4 | 98.8 | **+0.4** |
| Camera Viewpoints | 98.0 | 98.5 | **+0.5** |
| Language Instructions | 85.9 | 87.0 | **+1.1** |
| Light Conditions | 97.0 | 98.7 | **+1.7** |
| Objects Layout | 90.1 | 86.4 | -3.7 |
| Robot Initial States | 60.1 | 53.5 | -6.6 |
| Sensor Noise | 98.6 | 98.8 | **+0.2** |
| *Panel B: By Task Difficulty Level* | | | |
| Level 1 (Easiest) | 95.1 | 93.9 | -1.2 |
| Level 2 | 94.3 | 95.9 | +1.6 |
| Level 3 | 91.9 | 90.7 | -1.2 |
| Level 4 | 86.7 | 85.2 | -1.5 |
| Level 5 (Hardest) | 80.0 | 77.0 | -3.0 |
| **Overall** | **89.0** | **87.9** | **-1.1** |

*(c) LIBERO-Goal*

| Metric Category | $\pi_{0.5}$ Baseline | $\pi_{0.5}$ + GridS$_{32}$ | $\Delta$ |
|---|---|---|---|
| *Panel A: By Perturbation Dimension* | | | |
| Background Textures | 95.4 | 97.2 | **+1.8** |
| Camera Viewpoints | 93.9 | 96.1 | **+2.2** |
| Language Instructions | 67.8 | 64.4 | -3.4 |
| Light Conditions | 96.4 | 96.1 | -0.3 |
| Objects Layout | 73.2 | 64.2 | -9.0 |
| Robot Initial States | 78.0 | 75.3 | -2.7 |
| Sensor Noise | 94.2 | 95.3 | **+1.1** |
| *Panel B: By Task Difficulty Level* | | | |
| Level 1 (Easiest) | 94.7 | 95.0 | +0.3 |
| Level 2 | 91.6 | 92.8 | +1.2 |
| Level 3 | 91.3 | 89.0 | -2.3 |
| Level 4 | 82.0 | 80.7 | -1.3 |
| Level 5 (Hardest) | 63.5 | 57.6 | -5.9 |
| Unknown | 96.7 | 94.2 | -2.5 |
| **Overall** | **84.3** | **82.6** | **-1.7** |

*(d) LIBERO-Long (10)*

| Metric Category | $\pi_{0.5}$ Baseline | $\pi_{0.5}$ + GridS$_{32}$ | $\Delta$ |
|---|---|---|---|
| *Panel A: By Perturbation Dimension* | | | |
| Background Textures | 92.7 | 89.6 | -3.1 |
| Camera Viewpoints | 88.5 | 90.7 | **+2.2** |
| Language Instructions | 82.8 | 74.2 | -8.6 |
| Light Conditions | 92.0 | 95.6 | **+3.6** |
| Objects Layout | 86.5 | 81.4 | -5.1 |
| Robot Initial States | 65.4 | 55.7 | -9.7 |
| Sensor Noise | 88.4 | 88.6 | **+0.2** |
| *Panel B: By Task Difficulty Level* | | | |
| Level 1 (Easiest) | 90.5 | 84.3 | -6.2 |
| Level 2 | 92.2 | 88.0 | -4.2 |
| Level 3 | 87.6 | 83.1 | -4.5 |
| Level 4 | 83.7 | 81.5 | -2.2 |
| Level 5 (Hardest) | 75.5 | 75.3 | -0.2 |
| **Overall** | **84.6** | **81.6** | **-3.0** |

*consistently outperforms* the dense baseline across dimensions with extreme visual variance:

- *Camera Viewpoints & Light Conditions:* GridS achieves strictly superior performance in Camera Viewpoints (+3.0% in Spatial, +2.2% in Goal/10) and Light Conditions (+3.7% in Spatial, +3.6% in LIBERO-10).

- *Sensor Noise:* GridS also systematically outperforms the baseline under various sensor noise conditions (e.g., +1.7% in Spatial, +1.1% in Goal).

This phenomenon corroborates that the Information Bottleneck enforced by GridS is not merely an OOD generalization trick; it actively distills the representation during training. By aggressively pruning dense visual grids, GridS inherently prevents the transformer from assigning attention weights to unpredictable backgrounds, shadow artifacts, or noise pixels. Instead, it compels the model to reliably track robust, invariant causal geometries (such as the end-effector and target object contours) even within the training distribution.

**3. Analyzing the Resolution Trade-off (Where GridS Degrades).** The slight average performance drop (e.g., -1.2% in Spatial) can be directly attributed to dimensions that necessitate extremely high-resolution, fine-grained visual details:

- *Robot Initial States:* GridS exhibits performance drops (-2.7% to -9.7%) when the initial joint configuration is heavily randomized. To plan complex recovery trajectories from awkward initial poses, the model requires full-body kinematic awareness. Compressing the visual input to 32 tokens slightly over-smooths the proprioceptive boundary of the robot arm.

- *Objects Layout & Language Instructions:* GridS also experiences slight degradations (-2.1% to -9.0%) when the scene is cluttered with confounding objects or when language instructions require discriminating subtle textural differences. A strict token budget inherently sacrifices the fine-grained semantic resolution needed to perfectly ground highly complex linguistic queries against tiny distractors.

**Conclusion: Efficiency Meets Distillation.** In summary, the in-domain LIBERO-PLUS results complete the theoretical validation of GridS. While an 87.5% reduction in token density predictably entails a marginal cost in ultra-fine-grained kinematic and semantic resolution, it simultaneously functions as an innate spatial regularizer. By systematically outperforming the fully-augmented dense baseline in Camera, Noise, and Light perturbations, GridS cements its status as a paradigm that bridges massive computational acceleration with superior visual feature distillation.

E.1.3. ZERO-SHOT OUT-OF-DISTRIBUTION EVALUATION ON LIBERO-PLUS

To rigorously evaluate the generalization capabilities of our proposed GridS mechanism, we conduct zero-shot out-of-distribution (OOD) evaluations on the LIBERO-PLUS benchmark. We test the $\pi_{0.5}$ models (the dense baseline and the GridS variant with $K = 32$) across seven perturbation dimensions and five task difficulty levels (Level-1 to Level-5). The quantitative results across the four suites are detailed in Table 9.

**1. Difficulty Dynamics: The "Learning is Forgetting" Signature.** A granular analysis of the task difficulty levels (Panel B) reveals a profound dynamic. In highly predictable, easy scenarios (Level-1 and Level-2), the dense baseline slightly outperforms GridS because its 256 tokens enable perfect memorization of specific visual layouts. However, as the perturbation intensity reaches extreme limits (Level-5), the baseline's dense representation becomes a liability, leading to catastrophic degradation (e.g., plunging to 47.9% in LIBERO-Goal and 52.6% in LIBERO-10). In stark contrast, GridS substantially closes the gap and even overtakes the baseline at Level-5 (+0.5% in Goal, +3.4% in LIBERO-10). This confirms that compressing the visual representation acts as a strict information bottleneck: by "forgetting" task-irrelevant environmental noise, GridS prevents spurious overfitting and maintains causal robustness in extreme OOD conditions.

**2. Where GridS Excels: Visual and Spatial Robustness.** GridS demonstrates overwhelming superiority in perturbation dimensions that heavily distort visual geometry and rendering (Panel A).

- *Camera Viewpoints:* The dense baseline completely collapses under camera shifts (dropping to 67.0% in Spatial and 45.8% in LIBERO-10) due to severe *camera overfitting*—it relies on absolute pixel coordinates and background references. GridS actively discards the background and samples only relative, object-centric geometries, yielding massive inverse gains (+19.4% in Spatial, +10.8% in LIBERO-10).

*Table 9.* Zero-shot Out-of-Distribution Evaluation on the LIBERO-PLUS Benchmark. The $\pi_{0.5}$ baseline and the compressed model ($\pi_{0.5}$ + GridS$_{32}$) are evaluated under various perturbation dimensions (Panel A) and difficulty levels (Panel B). $\Delta$ denotes the absolute performance difference.

*(a) LIBERO-Spatial*

| Metric Category | $\pi_{0.5}$ **Baseline** | $\pi_{0.5}$ + **GridS$_{32}$** | $\Delta$ |
|---|---|---|---|
| *Panel A: By Perturbation Dimension* | | | |
| Background Textures | 97.3 | 95.0 | -2.3 |
| Camera Viewpoints | 67.0 | 86.4 | **+19.4** |
| Language Instructions | 92.6 | 82.6 | -10.0 |
| Light Conditions | 96.9 | 98.6 | +1.7 |
| Objects Layout | 96.9 | 95.1 | -1.8 |
| Robot Initial States | 83.4 | 72.6 | -10.8 |
| Sensor Noise | 90.9 | 91.5 | +0.6 |
| *Panel B: By Task Difficulty Level* | | | |
| Level 1 (Easiest) | 93.5 | 92.5 | -1.0 |
| Level 2 | 90.9 | 91.0 | +0.1 |
| Level 3 | 87.6 | 89.2 | +1.6 |
| Level 4 | 88.7 | 87.3 | -1.4 |
| Level 5 (Hardest) | 72.8 | 67.5 | -5.3 |
| **Overall** | **88.7** | **88.3** | **-0.4** |

*(b) LIBERO-Object*

| Metric Category | $\pi_{0.5}$ **Baseline** | $\pi_{0.5}$ + **GridS$_{32}$** | $\Delta$ |
|---|---|---|---|
| *Panel A: By Perturbation Dimension* | | | |
| Background Textures | 99.6 | 98.0 | -1.6 |
| Camera Viewpoints | 81.3 | 82.8 | +1.5 |
| Language Instructions | 92.4 | 80.8 | -11.6 |
| Light Conditions | 99.0 | 97.6 | -1.4 |
| Objects Layout | 90.1 | 87.6 | -2.5 |
| Robot Initial States | 73.6 | 61.8 | -11.8 |
| Sensor Noise | 95.5 | 97.9 | +2.4 |
| *Panel B: By Task Difficulty Level* | | | |
| Level 1 (Easiest) | 97.3 | 93.9 | -3.4 |
| Level 2 | 96.7 | 92.8 | -3.9 |
| Level 3 | 94.7 | 90.3 | -4.4 |
| Level 4 | 86.1 | 83.4 | -2.7 |
| Level 5 (Hardest) | 75.9 | 72.4 | -3.5 |
| **Overall** | **89.3** | **85.7** | **-3.6** |

*(c) LIBERO-Goal*

| Metric Category | $\pi_{0.5}$ **Baseline** | $\pi_{0.5}$ + **GridS$_{32}$** | $\Delta$ |
|---|---|---|---|
| *Panel A: By Perturbation Dimension* | | | |
| Background Textures | 87.9 | 95.0 | +7.1 |
| Camera Viewpoints | 71.3 | 80.9 | **+9.6** |
| Language Instructions | 72.0 | 60.0 | -12.0 |
| Light Conditions | 85.7 | 97.8 | **+12.1** |
| Objects Layout | 70.1 | 65.9 | -4.2 |
| Robot Initial States | 78.5 | 69.7 | -8.8 |
| Sensor Noise | 88.7 | 90.0 | +1.3 |
| *Panel B: By Task Difficulty Level* | | | |
| Level 1 (Easiest) | 95.5 | 93.5 | -2.0 |
| Level 2 | 91.1 | 92.5 | +1.4 |
| Level 3 | 87.1 | 85.6 | -1.5 |
| Level 4 | 74.2 | 70.9 | -3.3 |
| Level 5 (Hardest) | 47.9 | 48.4 | **+0.5** |
| Unknown | 86.0 | 97.5 | **+11.5** |
| **Overall** | **78.2** | **78.0** | **-0.2** |

*(d) LIBERO-Long (10)*

| Metric Category | $\pi_{0.5}$ **Baseline** | $\pi_{0.5}$ + **GridS$_{32}$** | $\Delta$ |
|---|---|---|---|
| *Panel A: By Perturbation Dimension* | | | |
| Background Textures | 93.8 | 84.1 | -9.7 |
| Camera Viewpoints | 45.8 | 56.6 | **+10.8** |
| Language Instructions | 92.2 | 85.6 | -6.6 |
| Light Conditions | 91.6 | 87.6 | -4.0 |
| Objects Layout | 92.0 | 84.0 | -8.0 |
| Robot Initial States | 78.9 | 64.9 | -14.0 |
| Sensor Noise | 78.4 | 84.0 | +5.6 |
| *Panel B: By Task Difficulty Level* | | | |
| Level 1 (Easiest) | 95.0 | 91.0 | -4.0 |
| Level 2 | 95.6 | 88.4 | -7.2 |
| Level 3 | 91.3 | 83.7 | -7.6 |
| Level 4 | 83.0 | 80.1 | -2.9 |
| Level 5 (Hardest) | 52.6 | 56.0 | **+3.4** |
| **Overall** | **80.0** | **77.1** | **-2.9** |

- *Sensor Noise & Light Conditions:* Under heavy noise (e.g., motion/Gaussian blur) and drastic illumination changes, GridS consistently outperforms the baseline (+5.6% Noise in LIBERO-10; +12.1% Light in Goal). The sparse sampling of 32 keypoints inherently bypasses high-frequency pixel corruption, allowing the policy to focus solely on uncorrupted topological affordances.

**3. Where GridS Struggles: Fine-Grained Kinematics and Semantics.** While highly resilient to visual variance, extreme compression inevitably introduces specific trade-offs:

- *Robot Initial States:* GridS exhibits performance drops (-8.8% to -14.0%) when the robot's initial joint angles are heavily perturbed. In difficult instances where the arm starts far from the training distribution, complex recovery trajectory planning requires high-resolution visual feedback of the entire kinematic chain. Aggressively compressing the scene down to 32 tokens over-smooths these fine-grained proprioceptive details.

- *Language Instructions:* We observe a moderate degradation (-6.6% to -12.0%) under language rewriting. Because GridS operates by aggressively filtering visual clutter, it may inadvertently discard subtle visual cues (e.g., slight textural variations or small contextual objects) that are strictly necessary to ground and decode highly complex, reasoned language instructions.

**4. Superiority over Discrete Pruning Paradigms and Final Advantage.** To contextualize these results, it is imperative to note that existing discrete token reduction methods, such as FastV and SparseVLM, *suffer catastrophic performance drops exceeding 20% on the LIBERO-PLUS dataset.* By discretely dropping patches, they fracture the continuous 2D spatial structure; when subjected to viewpoint shifts, these quantization errors are fatally magnified. GridS fundamentally resolves this via Differentiable Bilinear Sampling.

Ultimately, maintaining near-lossless overall performance (-0.4% in Spatial, -0.2% in Goal) while compressing the visual input by $87.5\%$ ($256 \rightarrow 32$ tokens) is a monumental achievement. GridS proves to be far more than an inference accelerator— it is a powerful representation distillator. Its true advantage over the dense baseline lies in its ability to break the fragility of dense vision, trading negligible absolute fidelity for breakthrough spatial and visual robustness.

### E.2. RoboTwin

During the rebuttal period, we received a suggestion from Reviewer LrTB to conduct testing on RoboTwin 2.0 (Chen et al., 2025). Therefore, we trained and deployed using the X-VLA method (Zheng et al., 2025) within a limited time. We conducted new experiments on the complex "Place Bread Skillet" task. This suite is explicitly designed to evaluate long-horizon planning and sequential multi-stage execution. As shown in our new results, GridS effectively maintains geometric attention over extended horizons, vastly outperforming the full-token baseline: while the fine-tuned standard XVLA [1] achieves only a 12% success rate, our XVLA+GridS policy yields a remarkable 76% success rate. We will add more result of RoboTwin 2.0 on the next period of discussion. Furthermore, our existing real-world stacking task is inherently a multi-stage process (approach $\rightarrow$ grasp $\rightarrow$ align $\rightarrow$ stack) requiring sustained spatial reasoning. GridS achieved a massive +52.4% success rate improvement here over the baseline.

Together, these results clearly demonstrate that GridS does not degrade in challenging settings; rather, it robustly supports rich, long-horizon interactions by preventing spatial error accumulation. We will include these new RoboTwin 2.0 results in the revised manuscript.

## F. Evaluation of Alternative Downsampling Baselines

To rigorously isolate the contribution of our proposed geometry-aware sampling mechanism from the mere reduction in sequence length, we evaluate GridS against two alternative downsampling baselines:

- **Random Sampling:** Randomly selecting $K$ tokens from the dense visual feature grid.

- **Top-$K$ Pruning:** Discretely selecting the top $K$ tokens based on visual saliency (e.g., maximum activation norms).

For a strictly fair comparison, all variants are restricted to an identical token budget ($K = 16$) and are trained under the exact same full fine-tuning protocol on the LIBERO dataset. The quantitative comparisons are reported in Table 10.

*Table 10.* Performance comparison of different downsampling mechanisms under a matched fine-tuning protocol on the LIBERO dataset. All compressed variants use exactly 16 visual tokens.

| Method | Tokens | Spatial | Object | Goal | Long | Avg. SR (%) |
|---|---|---|---|---|---|---|
| Baseline ($\pi_0$) | 256 | 97.2 | 98.8 | 96.0 | 85.6 | 94.4 |
| Random Sampling | 16 | 93.8 | 98.4 | 85.8 | 73.2 | 87.8 |
| Top-$K$ Pruning | 16 | 96.6 | 98.8 | 89.8 | 77.0 | 90.5 |
| **GridS (Ours)** | **16** | **98.0** | **99.2** | **96.4** | **90.2** | **96.0** |

**Analysis:** As shown in Table 10, naive token reduction strategies fail to maintain the baseline's capabilities. Random Sampling drastically degrades the average success rate to 87.8%, with a particularly severe collapse in long-horizon tasks (73.2%), as it arbitrarily discards critical interactive geometries. Top-$K$ Pruning performs marginally better (90.5%) by prioritizing salient features, but still falls significantly short of the dense baseline. This performance gap highlights the fundamental flaw of discrete patch dropping: it destroys continuous 2D spatial structures, resulting in geometric quantization errors that severely impair precise motor control.

In contrast, GridS not only recovers this performance drop but actively outperforms the 256-token baseline. These results conclusively demonstrate that the performance gains of GridS are not simply a byproduct of reducing the token count (which might otherwise prevent overfitting), but are intrinsically derived from its differentiable, geometry-aware resampling mechanism that effectively filters noise while preserving essential physical precision.

## G. Extreme Compression: A Novel Information Bottleneck Paradigm

To push the limits of our proposed sampling mechanism, we conduct an extreme stress test on the LIBERO dataset by restricting the GridS budget to a single visual token ($K = 1$). We employ the fine-tuned $\pi_{0.5}$ architecture for this ablation to verify if the mechanism holds under a highly capable backbone. The results of this ultra-sparse configuration are reported in Table 11.

*Table 11.* Performance of GridS under the extreme sparsity constraint ($K = 1$) using the $\pi_{0.5}$ backbone. Counter-intuitively, a single sampled token (a 99.6% reduction in visual input) retains sufficient information to achieve nearly identical performance compared to the dense 256-token baseline.

| Model Config | Spatial | Object | Goal | Long | Avg. SR (%) |
|---|---|---|---|---|---|
| $\pi_{0.5}$ Baseline (256 Tokens) | 98.4 | 98.0 | 97.6 | 92.8 | 96.7 |
| $\pi_{0.5}$ **+ GridS** ($K = 1$) | 98.0 | 99.0 | 97.6 | 91.8 | 96.6 |

**Analysis from an Information Retention Perspective:**

The astonishing capability of a single-token representation (averaging 96.6%, practically on par with the 96.7% of the dense $\pi_{0.5}$ baseline) provides profound insight into visual redundancy in robotic manipulation. From an information theory perspective, this phenomenon can be explained through the *Information Bottleneck (IB)* principle.

In standard dense grid representations, the model is fed all visual information indiscriminately. While this mathematically maximizes "information retention", it simultaneously introduces massive amounts of task-irrelevant background noise and domain-specific textures. This overabundance of redundant capacity allows the over-parameterized VLA policy to overfit to spurious spatial correlations—essentially memorizing the dataset layout rather than learning generalizable interaction skills.

By imposing an extreme constraint ($K = 1$), GridS creates a strict information bottleneck. To minimize task loss under this severe limitation, the coordinate predictor is forced to discard all redundant background context and actively map the single sampling coordinate to the absolute semantic center of the interactive scene (e.g., the precise affordance point between the gripper and the target).

**A Novel Compression Paradigm:**

These findings suggest that GridS operates fundamentally differently from traditional token pruning. Traditional methods attempt to passively "preserve what is important" by selecting multiple discrete patches. In contrast, GridS introduces an

active representation distillation paradigm. By drastically reducing the volume of training information (a 99.6% compression rate with negligible performance degradation), the differentiable bottleneck inherently prevents the model from relying on spatial shortcuts. Instead, it compels the downstream Transformer to learn the true causal physical mechanics (the essential geometry and affordances) required to complete the task. This paradigm shifts the focus of VLA perception from "maximizing input coverage" to "extracting minimal sufficient statistics", opening a new pathway for highly generalized and computationally minimal robotic learning.

## H. Ablation Studies on GridS Architecture

To thoroughly evaluate the architectural design choices of GridS, we conduct a comprehensive ablation study on the LIBERO dataset. We systematically isolate and modify key components, including the pooling mechanism, predictor capacity, coordinate injection, and interpolation strategy. The quantitative results are summarized in Table 12.

*Table 12.* Ablation study of GridS components on the LIBERO dataset. The default setting serves as the reference baseline.

| Component Changed | Variant | $\Delta$ Success Rate |
|---|---|---|
| *Default GridS* | (GAP, Base MLP, w/ Coord, Bilinear) | Reference |
| (a) Pooling Strategy | Conv-Downsample | -4.4% |
| | Global Max Pooling (GMP) | -0.3% |
| (b) Predictor Capacity | 10× Larger MLP | +0.2% |
| | 100× Larger MLP | +0.2% |
| | Segment Anything Model | -6.2% |
| (c) Coordinate Injection | w/o Coordinate Injection | -3.6% |
| (d) Interpolation Method | Nearest-Neighbor | -4.1% |

Based on the experimental results, we detail the analysis of each component below:

- **Pooling Strategy:** The parameter-free Global Average Pooling (GAP) serves as a robust global summarizer. Replacing it with a parameterized convolutional downsampling module leads to a 4.4% performance drop, largely due to overfitting to the spatial biases present in the dataset. Alternatively, Global Max Pooling (GMP) slightly degrades performance (-0.3%) by discarding continuous global spatial context.

- **Predictor Capacity:** Expanding the MLP predictor size by 10× or even 100× yields a negligible +0.2% improvement. This indicates that mapping global visual context to a sparse set of 2D coordinates is an inherently low-complexity task, where scaling up model capacity offers no meaningful benefit. We further experimented with utilizing a frozen, pre-trained Segment Anything Model (SAM) (Kirillov et al., 2023) to guide the coordinate sampling. Surprisingly, this led to a severe **-6.2%** performance drop. This strongly highlights that generic semantic segmentation (identifying *what* an object is) misaligns with task-driven geometric sampling (identifying exactly *where* the gripper should interact).

- **Coordinate Injection:** Removing explicit coordinate injection results in a 3.6% performance drop. Because the sparse continuous sampling inherently disrupts the original dense 2D grid structure, explicitly injecting spatial coordinates is essential for the downstream Transformer to retain accurate positional awareness of the objects.

- **Interpolation Method:** Substituting bilinear interpolation with nearest-neighbor sampling significantly degrades performance (-4.1%). The discrete rounding operation in nearest-neighbor sampling breaks the gradient flow. In contrast, bilinear interpolation preserves full differentiability, which is strictly required for end-to-end, task-driven optimization.

## I. Calculation of the Information Retention Map

To quantitatively evaluate the semantic coverage of the sampled tokens, we introduce the Information Retention map (as visualized in Figure 7).

For each original dense token $F_{\text{orig}}^{(i,j)} \in \mathbb{R}^C$ at spatial location $(i,j)$, we compute its maximum cosine similarity against all $K$ sampled sparse tokens $F_{\text{samp}}^{(k)} \in \mathbb{R}^C$. The retention score $R$ at location $(i,j)$ is mathematically defined as:

$$R(i,j) = \max_{k \in \{1,\dots,K\}} \left( \frac{F_{\text{orig}}^{(i,j)} \cdot F_{\text{samp}}^{(k)}}{\|F_{\text{orig}}^{(i,j)}\|\|F_{\text{samp}}^{(k)}\|} \right) \tag{5}$$

Intuitively, a high retention score (approaching $1.0$, represented by the green regions in our visualizations) indicates that the semantic information of that specific original patch is successfully preserved by at least one token in the aggressively compressed GridS set.

## J. Limitations

While GridS demonstrates strong empirical performance and theoretical efficiency, we identify two primary limitations in our current framework.

First, the practical wall-clock speedup during single-batch inference is relatively modest (approximately $1.2\times$) despite massive FLOPs reductions. Because dense baseline models are already heavily optimized via JAX compilation, the runtime bottleneck during sequential, single-batch execution shifts from active computation to fixed kernel overheads. Consequently, the maximum efficiency benefits of GridS are currently realized primarily in high-throughput, large-batch settings.

Second, the sampling mechanism relies on a static, predefined token budget (e.g., $K = 16$) across all tasks. Our ablation studies reveal an inverted U-shaped performance trend regarding token density, indicating that an excessively large $K$ degrades performance by reintroducing irrelevant background noise in precision-sensitive scenarios. Currently, GridS cannot adaptively adjust this token budget based on real-time scene complexity. Extending the architecture to dynamically predict the optimal token count $K$ per frame represents a critical direction for future research.

Third, GridS currently exhibits sub-optimal compatibility with Parameter-Efficient Fine-Tuning (PEFT) methods (Houlsby et al., 2019). Our experiments indicate that when adapted using LoRA (Hu et al., 2022) rather than full fine-tuning, GridS underperforms the dense baseline by 8.3%. We hypothesize that this stems from a severe representational mismatch: GridS fundamentally alters the spatial distribution of visual tokens (shifting from dense, uniform grids to sparse, continuously interpolated features). While full fine-tuning provides sufficient parameter capacity to realign the VLA's attention mechanism to this novel input structure, LoRA's restricted degrees of freedom struggle to overcome the frozen pre-trained dense priors. Addressing this capacity bottleneck to enable highly efficient PEFT deployment is an important next step.

## K. Additional Visualizations of Information Retention Maps

To further demonstrate the robust and consistent behavior of the GridS sampling mechanism, we provide additional visualizations of the Information Retention maps evaluated on the LIBERO benchmark (Figure 10) and ALOHA (Figure 11). In addition, we provide the full sequence of Information Retention maps for the real-world "Stacking Cubes" task (Steps 0–47) in Figure 12.

An important empirical observation is that during successful real-world execution, the average retention score fluctuates consistently between $0.8$ and $0.9$, rather than maintaining near-perfect $1.0$ fidelity. As analyzed below, this "lossy" compression is a key driver of our model's superior out-of-distribution (OOD) performance.

The observation that intermediate retention ($0.8 \sim 0.9$) outperforms full retention ($1.0$) aligns with the "Learning is Forgetting" principle (Conklin et al., 2026). In VLA models, perfect information retention (as seen in the dense baseline) often leads to the inclusion of high-frequency noise and task-irrelevant environmental features. This over-abundance of information allows the model to find "spatial shortcuts" or overfit to spurious correlations within the training distribution. In contrast, GridS introduces a principled Information Bottleneck. By restricting the retention to approximately $85\%$, the model is forced to "forget" non-causal background pixels. This compression compels the Transformer backbone to rely exclusively on the invariant, minimal sufficient statistics of the scene, namely the physical affordances and object-effector geometries. Consequently, GridS achieves significantly higher success rates in real-world OOD scenarios where the baseline's "perfect memory" of training-specific noise becomes a liability.

Viewed through the lens of information retention, a score approaching $1.0$ indicates high reconstructive fidelity, yielding

performance that closely mirrors the dense baseline. Conversely, an intermediate retention regime of $0.8 \sim 0.9$ acts as a critical information bottleneck. By intentionally discarding $10\% \sim 20\%$ of the peripheral visual signals, this lossy compression compels the model to 'forget' spurious background noise and distill the true causal features governing the manipulation task.

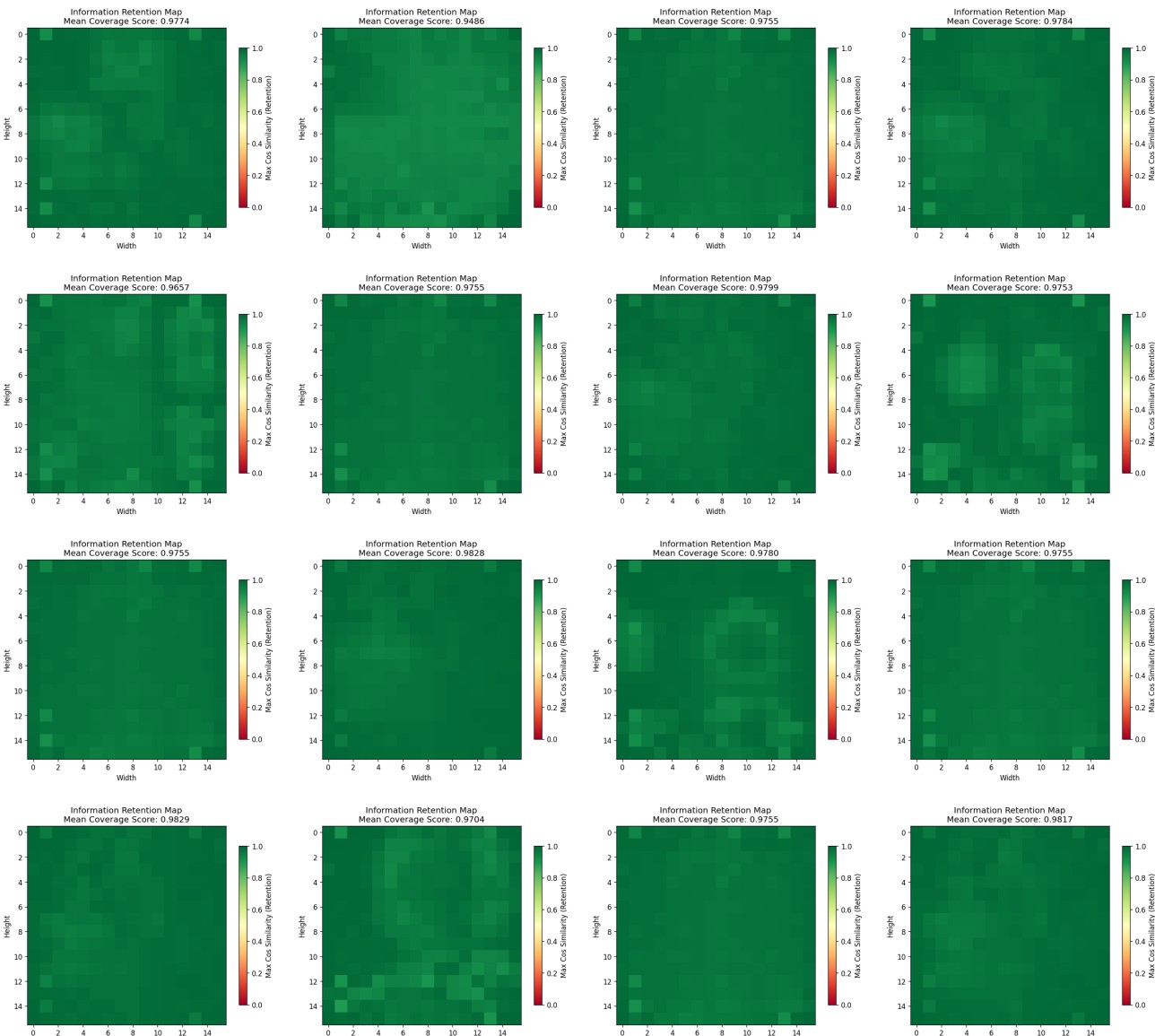

*Figure 10.* Additional Information Retention maps on the LIBERO dataset.

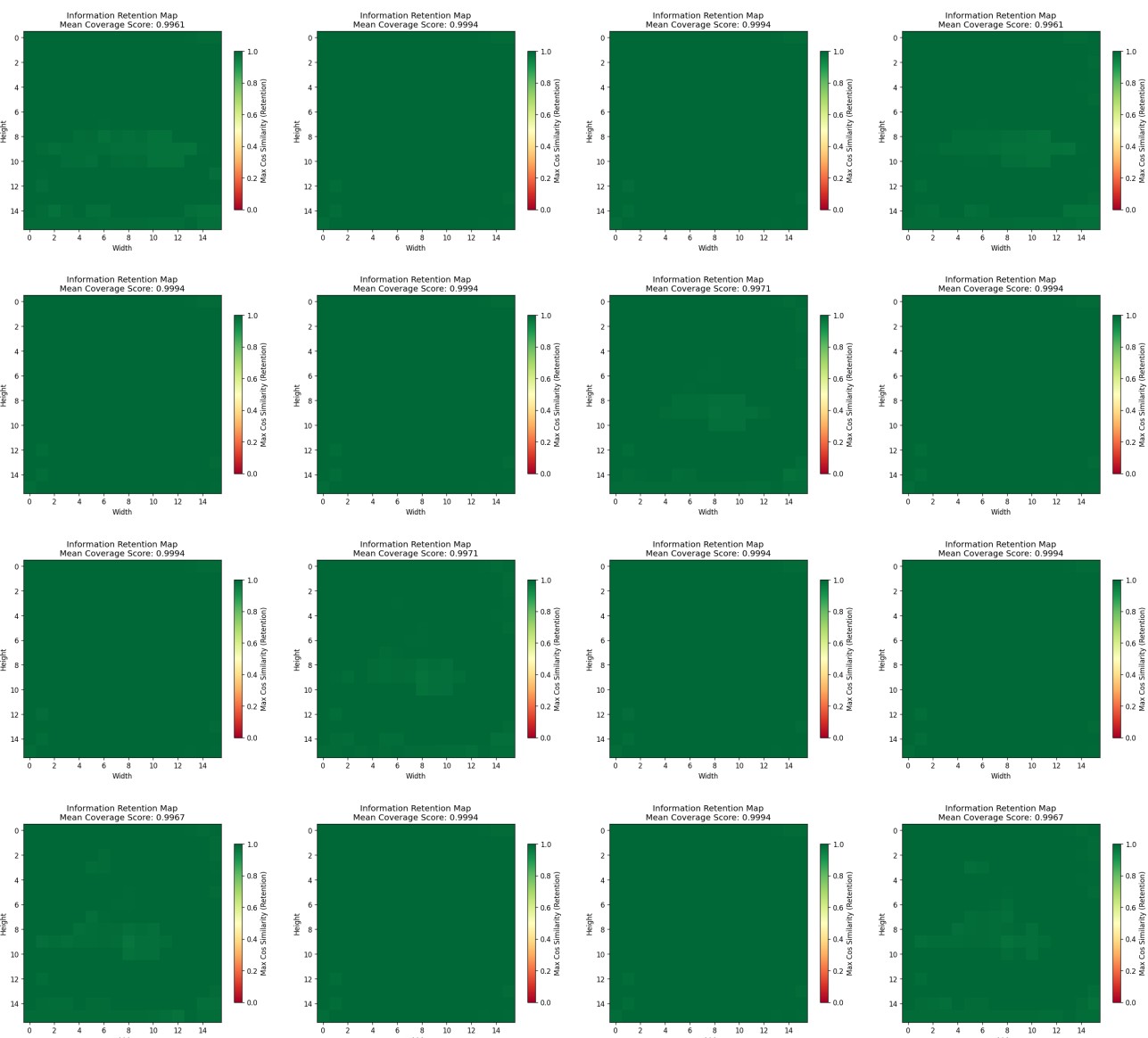

*Figure 11.* Additional Information Retention maps on the ALOHA dataset.

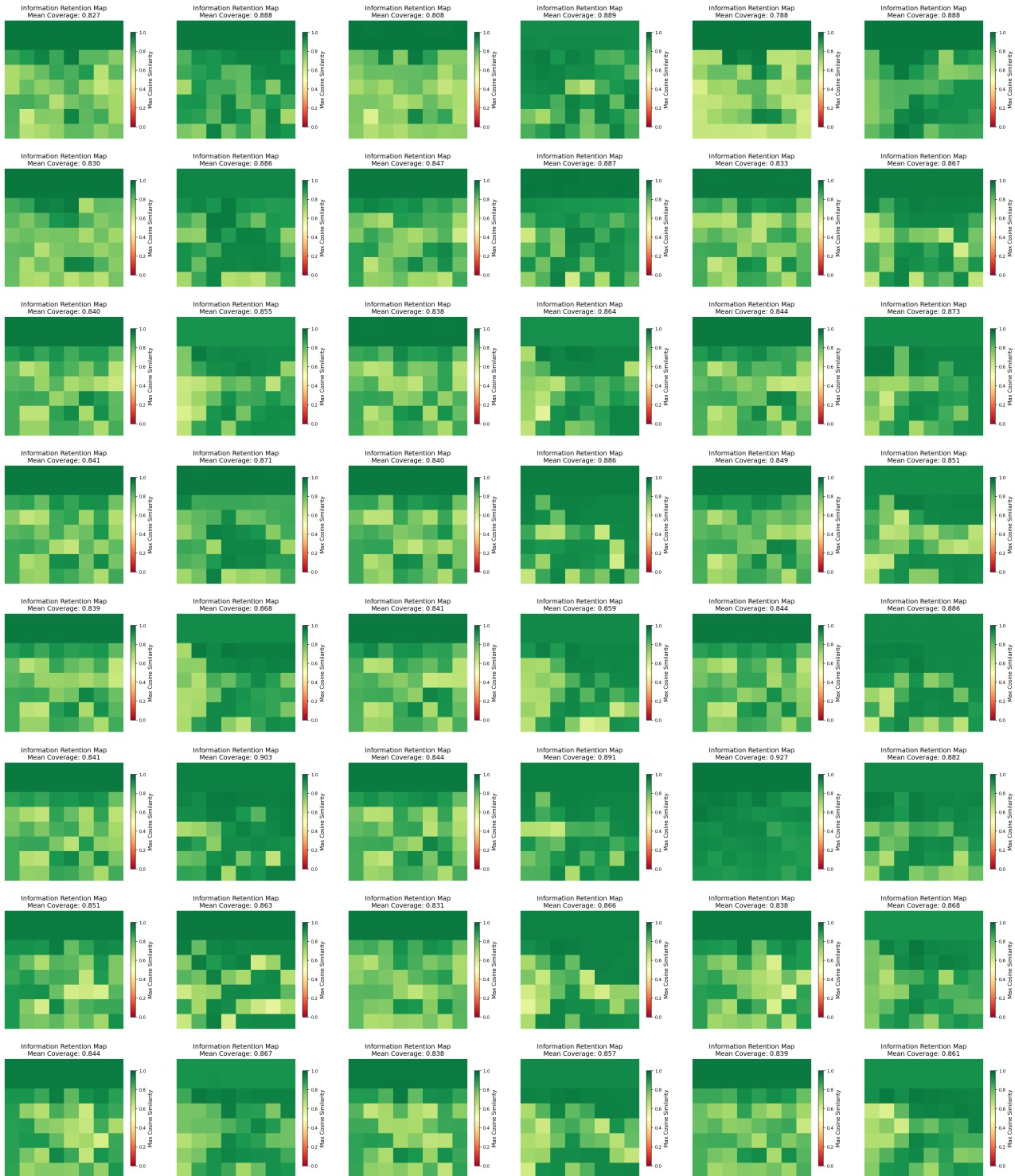

*Figure 12.* Continuous Information Retention maps for the Real-World Stacking task (Steps 0–47). The visualization demonstrates that the model consistently maintains a retention score of $0.8 \sim 0.9$, effectively filtering background distractors while focusing on the relative geometry between the gripper and cubes.

