# OpenReview forum: "See What Matters: Differentiable Grid Sample Pruning for Generalizable Vision-Language-Action Model"
_ICML.cc/2026/Conference — ICML 2026 regular_

### Official Review · Reviewer_741F · 2026-03-06

**Soundness:** 3
**Presentation:** 3
**Significance:** 3
**Originality:** 3
**Overall Recommendation:** 4
**Confidence:** 3

**Summary:**

In order to improve performance and computational efficiency of VLAs, this paper proposes using a relatively small number of visual features sampled via bilinear interpolation from the output of a vision encoder. Since the sampling procedure is differentiable, the coordinates of the sampled features can be learned end-to-end, and are dynamically predicted with a small network based on pooled vision encoder features. Experiments demonstrate that this simple sampling-based feature extraction procedure decreases the computational cost while maintaining or even improving accuracy and success rates on LIBERO and ALOHA benchmarks, as well as a smaller scale real-world experiment.

**Compliance With Llm Reviewing Policy:**

Affirmed.

**Final Justification:**

My original assessment already found the work to be simple/elegant and likely effective, but I could not recommend acceptance due to significant omissions from the evaluation. During the discussion period, the authors have run additional experiments to more conclusively demonstrate the effectiveness of their proposed approach. With these additional experiments, the paper is now much more compelling.

**Key Questions For Authors:**

* **Q1**: What is the training/fine-tuning protocol used to adapt GridS to an existing VLA model? Do you only train the sampling coordinate prediction MLP, or do you also fine-tune the VLA model?
* **Q2**: Can you provide more details on the computation of the “information retention map” from Figure 7? Is it possible to compute similar metrics/visualizations for other token pruning methods/baselines?
* **Q3**: Various design choices in GridS, related to the following components, were not experimentally justified:
  * **a.** The coordinate prediction feature network’s input feature: you use global average pooling, but what about other pooling or feature extraction methods?
  * **b.** The coordinate prediction network itself: you use a lightweight MLP, does its size have any significant impact on performance?
  * **c.** The injection of coordinate information from the coordinate encoder: how important is it? I did not see any ablation of this component.
  * **d.** Can the performance gains mainly be attributed to learning the position of tokens to sample in an end-to-end fashion, or is the bilinear interpolation also crucial? An experiment replacing bilinear interpolation with nearest-neighbor interpolation at test-time could help answer this question.

**Limitations:**

Limitations are not discussed. It may be interesting to mention limitations related to the need of fine-tuning downstream models (if this is the case), or the use of a fixed number of tokens regardless of scene or task complexity.

**Strengths And Weaknesses:**

The simplicity and elegance of the method are major strengths of the proposed approach, and I found the paper to be well written and easy to understand. Thanks to the method’s simplicity, it has the potential to be broadly applied as part of many pipelines requiring visual feature extraction. While straightforward in hindsight, I have also not seen a grid-sampling based approach for sparse feature extraction in the context of VLMs/VLA models.

My main concern is related to the soundness of experiments comparing GridS to other pruning methods: while experiments show that GridS is beneficial when applied to the $\pi_0$ and $\pi_{0.5}$ VLAs, comparisons with respect to prior pruning methods (FastV, SparseVLM, VLA-Cache) would be fairer if GridS was applied to the same OpenVLA-OFT model as those baselines. Alternatively, applying prior pruning methods to the $\pi_0$ and $\pi_{0.5}$ models would also allow for fairer comparison. As it stands, it is not possible to conclusively say that GridS would outperform those methods when applied to OpenVLA-OFT. Conversely, it is also hard to conclude that GridS would outperform other pruning approaches when applied to $\pi_0$/$\pi_{0.5}$.

It was also not clear to me whether $\pi_0$/$\pi_{0.5}$ are fine-tuned together with the coordinate prediction MLP, or whether they can remain frozen. If they are fine-tuned, it would be important to note this distinction when comparing to some other pruning methods which may not need any fine-tuning of the downstream VLA model.

I would also have appreciated to see a few more experiments related to exploring the design space of the GridS method. Without such experiments, the paper leaves a few questions related to the importance of various design choices unanswered. See Q3 under “Key Questions” for specific examples of design choices that could be valuable to discuss.

---

> ### Author Rebuttal · Authors · 2026-03-31
>
> We sincerely thank Reviewer 741f for appreciating the simplicity, elegance, and potential broad applicability of GridS.
>
> > **W1/W2/Q1: Experimental fairness across base models & Training protocol.**
>
> We respectfully argue that being "training-free" is not an inherent advantage for these prior methods, but rather a forced compromise due to their non-differentiable designs. To avoid ambiguity, we will add the following content to the experimental section of the main text.
>
> VLA typically involves a two-stage training pipeline: **1)** a lengthy pre-training phase, followed by **2)** full fine-tuning or LoRA adaptation for downstream tasks. Existing training-free pruning methods (e.g. FastV, SparseVLM) are applied after both stages, i.e., at test time, which essentially introduces an additional step with limited effectiveness. The cost of this compromise is the performance degradation observed when strictly evaluated on downstream tasks (as shown in Table 1, where these methods drop performance by -0.2% to -2.0%).
>
> In contrast, GridS is integrated directly between the backbone’s vision encoder and transformer during the second stage. By leveraging differentiability, it performs pruning jointly with the backbone during full fine-tuning, a step that $\pi_0$ already requires irrespective of GridS. This differentiability enables the pruning mechanism to be jointly optimized with the task loss, effectively solving the performance degradation problem that plagues non-differentiable, training-free methods. As a result, GridS incurs no extra training overhead and, instead, significantly reduces training costs in the second stage (as shown in Table 4, GridS achieved a substantial 3.4× speedup for $\pi_0$ at a batch size of 128).
>
> VLA-Cache is an autoregressive-model-based pruning method and thus cannot be applied to $\pi_0$. GridS, on the other hand, requires full fine-tuning, making it incompatible with OpenVLA-OFT, which only support LoRA-based adaptation. Therefore, to strictly verify that our differentiable GridS is superior to training-free pruning methods, we applied both FastV and SparseVLM to our exact same $\pi_0$ architecture (all method are finetuning 30k iterations). The comparison is as follows:
>
> | Method | Tokens | Spatial | Object | Goal | Long | Avg. SR |
> | :--- | :---: | :---: | :---: | :---: | :---: | :---: |
> | $\pi_0$ | 256 | 97.2 | 98.8 | 96.0 | 85.6 | 94.4
> | $\pi_0$ + FastV | 100 | 97.0 | 98.4 | 93.8 | 82.4 | 92.9
> | $\pi_0$ + SparseVLM | 100 | 93.4 | 98.0 | 91.2 | 76.6 | 89.8
> | **$\pi_0$ + GridS (Ours)** | **4** | 96.6 | 99.4 | 96.4 |89.6 | **95.5**
>
> Under the same aggressive compression ratio applied to the same base model, non-differentiable methods suffer from significant performance degradation. In stark contrast, our trainable continuous sampling approach retains high accuracy, demonstrating that GridS does not merely drop tokens but actively learns to sample the most geometry-critical regions.
>
>
>
> > **Q2: Details on the "Information Retention map" in Fig 7.**
>
> We apologize for the lack of detail in the original manuscript.
> For each original dense token $F_{orig}^{(i,j)} \in \mathbb{R}^{C}$ at spatial location $(i,j)$, we compute its maximum cosine similarity against all $K$ sampled sparse tokens $F_{samp}^{(k)} \in \mathbb{R}^{C}$. The retention score $R$ at location $(i,j)$ is mathematically defined as:
>
> $$R(i,j) = \max_{k \in \{1 \dots K\}} \left( \frac{F_{orig}^{(i,j)} \cdot F_{samp}^{(k)}}{\|F_{orig}^{(i,j)}\| \|F_{samp}^{(k)}\|} \right)$$
>
> High values (green in Fig. 7) indicate that the original patch's semantics are successfully preserved by the compressed GridS tokens. We will add this formulation and comparative visualizations of different pruning strategies to the revision.
>
> > **Q3: GridS design choices are not justified by experiments.**
>
> We conducted a comprehensive ablation on the LIBERO dataset:
>
> | Component Changed | Variant | Success Rate on LIBERO |
> | :--- | :--- | :---: |
> | **Default GridS** | (GAP, Base MLP, w/ Coord, Bilinear) | **Reference** |
> | **(a) Pooling** | Conv-Downsample | -4.4%  |
> | | Global Max Pooling | -0.3% |
> | **(b) Predictor**| 10$\times$ Larger MLP | +0.2% |
> | | 100$\times$ Larger MLP | +0.2% |
> | **(c) Coord Encoder**| w/o Coordinate Injection | -3.6% |
> | **(d) Interpolation** | Nearest-Neighbor | -4.1% |
>
> * (a) Pooling: Parameter-free GAP prevents overfitting. Conv-Downsample severely overfits to spatial biases (-4.4%), while GMP loses global context.
> * (b) Predictor: A 100$\times$ larger MLP yields negligible gains (+0.2%), confirming the global-to-coordinate mapping is inherently low-complexity.
> * (c) Coord Encoder: Removing it (-3.6%) destroys spatial awareness, as explicit injection is required to locate the sparsely sampled tokens.
> * (d) Interpolation: Nearest-Neighbor's discrete operation breaks the gradient flow (-4.1%). Bilinear interpolation is essential for end-to-end differentiability.

---

> > ### Author Rebuttal · Reviewer_741F · 2026-04-02
> >
> > Thank you for your response, which has addressed my main concerns: fair evaluation using a uniform protocol across baselines, and justification of design choices via ablations. Your answer to W1/W2 of Reviewer 5e6E also helped to further demonstrate the value of learned sample location prediction as opposed to other downsampling techniques under a consistent training and evaluation protocol. Therefore, I will raise my score to 4.

---

> > > ### Author Response · Authors · 2026-04-02
> > >
> > > We sincerely thank the reviewer for reading our rebuttal and raising the score to a Weak Accept.
> > > We are very glad that the uniform baseline comparisons and ablation studies successfully addressed your concerns regarding experimental fairness.
> > > We will ensure that these strict evaluations and the related discussions are prominently included in the final camera-ready version.

---

### Official Review · Reviewer_5e6E · 2026-03-08

**Soundness:** 3
**Presentation:** 3
**Significance:** 2
**Originality:** 2
**Overall Recommendation:** 4
**Confidence:** 4

**Summary:**

This paper proposes a simple and plug-and-play approach to improve the efficiency of VLA models. The authors observe that VLA architectures often process a large number of visual tokens, while existing token pruning approaches typically rely on selecting discrete patches from a fixed grid, which may overlook sub-patch level geometric information. To address this issue, the authors propose predicting a small number of sub-patch coordinates using globally pooled visual features. The features at these coordinates are then obtained through bilinear interpolation from nearby patch features, and the coordinate information is added as positional encoding. This allows the model to compress the number of visual tokens and improve efficiency during both training and inference. The method is evaluated in both simulation and real-world experiments and shows certain improvements.

**Compliance With Llm Reviewing Policy:**

Affirmed.

**Final Justification:**

The rebuttal has addressed a substantial portion of my concerns. In particular, the authors provide additional comparisons under matched settings and include simple baselines such as random and top-k token selection. These results make it much more convincing that the performance gains are not solely due to reducing the number of tokens, but also come from the proposed adaptive sampling mechanism itself.

The additional ablations on the predictor design are also helpful. Beyond Global Average Pooling, alternative architectures are evaluated, which strengthens the justification of the design choices.

That said, I still have some remaining reservations. The practical inference latency improvement is relatively limited compared to the FLOPs reduction, and the explanation for the large OOD performance gain is still not fully supported by direct evidence, relying more on a plausible interpretation.

Overall, I find that the rebuttal significantly improves the clarity and credibility of the work, and the remaining issues are not critical enough to outweigh its contributions. Therefore, I am willing to increase my score to 4 (weak accept).

**Key Questions For Authors:**

see Weaknesses.

**Limitations:**

yes

**Strengths And Weaknesses:**

**Strengths**

1. The problem addressed in this work is important for embodied systems. The large number of visual tokens in VLA models significantly affects efficiency, and this issue has not been sufficiently explored in the current literature.

2. The proposed method is simple and intuitive. It first predicts K important points using global features, then samples nearby patch features via bilinear interpolation and adds positional encoding. The design is plug-and-play and can be easily integrated into existing VLA architectures.

3. The method demonstrates promising empirical results. It achieves improved efficiency while maintaining comparable performance, and the effectiveness is shown in both simulation and real-world experiments.

**Weaknesses**

1. The paper lacks comparisons with other efficient VLA methods under the same baseline (e.g., π0, π0.5) and training/testing settings. In particular, although the paper criticizes approaches that select tokens from a fixed grid (discrete patch/token pruning), it does not provide direct comparisons with such methods under identical settings, making it difficult to clearly assess the advantage of the proposed approach.

2. The paper does not clearly disentangle the effect of token reduction from the proposed efficiency mechanism. The improvements in performance and efficiency may largely come from reducing the number of tokens rather than from GridS itself. Therefore, comparisons with simple baselines such as random token sampling and top-k attention pruning would be necessary.

3. Although the method significantly reduces FLOPs, the improvement in actual inference latency is relatively limited (around 1.2×), which weakens the practical efficiency gain.

4. The coordinate prediction mechanism is relatively simple. The key sampling points are predicted directly from global pooled features, and the paper does not evaluate the prediction quality or compare with alternative prediction strategies.

5. In Table 3, the real-world Stack Blocks experiment shows that moderately reducing the number of tokens leads to a very large performance improvement, especially in OOD scenarios. Explaining this solely as removing background noise seems insufficient. In principle, more visual tokens should provide more information, and background noise alone may not explain such a large performance drop.

---

> ### Author Rebuttal · Authors · 2026-03-31
>
> We thank Reviewer 5e6E for recognizing the importance of token efficiency in VLAs and the plug‑and‑play nature of GridS.
>
> > **W1 & W2: Lack of same-baseline comparison & Confounding effect of token reduction.**
>
> We completely agree that ensuring a strict same-baseline comparison and decoupling the GridS mechanism from mere token reduction is critical.
>
> First, we respectfully argue that being "training-free" (e.g., FastV, SparseVLM) is not an inherent advantage, but rather a forced compromise due to non-differentiable designs. VLAs typically require a lengthy pre-training phase followed by full fine-tuning/LoRA. Training-free methods are forced to operate post-training at test time, leading to inherent performance degradation. In contrast, GridS is fully differentiable. It is integrated and jointly optimized with the backbone during the standard full fine-tuning stage that $\pi_0$ already requires. Thus, GridS incurs no extra training overhead; instead, it provides a substantial **3.4× training speedup** (Table 4).
>
> VLA-Cache is autoregressive-specific and cannot be applied to $\pi_0$. Meanwhile, GridS requires full fine-tuning, making it incompatible with LoRA-only baselines like OpenVLA-OFT. Therefore, to provide the fairest apples-to-apples comparison, we applied FastV, SparseVLM, Random Sampling, and Top-K Pruning directly onto our exact $\pi_0$ architecture. All models were fine-tuned for 30k iterations under identical settings:
>
> | Method | Tokens | Spatial | Object | Goal | Long | Avg. SR (%) |
> | :--- | :---: | :---: | :---: | :---: | :---: | :---: |
> | Baseline | 256 | 97.2 | 98.8 | 96.0 | 85.6 | 94.4 |
> | FastV | 100 | 97.0 | 98.4 | 93.8 | 82.4 | 92.9 |
> | SparseVLM | 100 | 93.4 | 98.0 | 91.2 | 76.6 | 89.8 |
> | Random Sampling | 16 | 93.8 | 98.4 | 85.8 | 73.2 | 87.8 |
> | Top-K | 16 | 96.6 | 98.8 | 89.8 | 77.0 | 90.5 |
> | Ours | 16 | 98.0 | 99.2 | 96.4 | 90.2 | **96.0** |
>
>
> > **W3: Limited practical latency improvement despite huge FLOP reduction.**
>
> While single-batch inference latency improves moderately (1.2$\times$) due to fixed JAX compilation overheads, single-batch speed is not the primary bottleneck for current VLAs. Instead, GridS resolves the true critical challenges: prohibitive training costs and real-world fragility.
>
> * Accelerated Training: By drastically reducing FLOPs, GridS speeds up end-to-end training by 3.4$\times$ (Table 4), directly solving the offline training bottleneck.
> * Enhanced Real-World Robustness: The massive FLOP reduction acts as a spatial regularizer, forcing the policy to focus on critical geometries and yielding a +52.4% success rate jump in the OOD Stack Cubes task.
> * High-Throughput Speedup: In offline batch processing, the inference speedup scales rapidly to 3.2$\times$ (Figure 6).
>
>
> > **W4: The coordinate prediction mechanism is too simple (GAP) and lacks alternative evaluation.**
>
> To explore this, we evaluated more complex alternatives on LIBERO (to be added in the Appendix) and found that increased model complexity does not yield meaningful improvements. Specifically, replacing GAP with a parameterized Conv-Downsample leads to overfitting and a 4.4% performance drop, while scaling the predictor MLP size by up to 100x provides a negligible +0.2% gain, confirming the inherently low-complexity nature of mapping global visual context to 2D coordinates. Furthermore, incorporating heavyweight visual priors via a frozen, pre-trained Segment Anything Model (SAM) surprisingly resulted in a severe 6.2% performance degradation. This strongly highlights that generic semantic segmentation (identifying what an object is) fundamentally misaligns with the precise, task-driven geometric sampling (identifying exactly where to interact) required for our method.
>
> > **W5: Insufficient explanation for the huge OOD performance jump.**
>
> We thank the reviewer for this insightful question. While "more tokens provide more information" applies to standard vision tasks, it exacerbates causal confusion [1, 2] in imitation learning. Dense visual inputs allow policies to exploit non-causal background features as shortcuts (spurious correlations). In OOD scenarios with novel layouts, these spurious correlations break and cause catastrophic compounding errors.
>
> GridS solves this by acting as a strict spatial regularizer. Aggressively compressing the visual input into exactly 16 geometric coordinates forcibly discards non-causal background noise. This prevents spurious correlations and forces the policy to rely exclusively on invariant, interaction-centric features. This theoretical mechanism explains why GridS prevents OOD degradation and achieves the massive +52.4% success rate jump in our real-world evaluations.
>
> [1] De Haan P, Jayaraman D, Levine S. Causal confusion in imitation learning[J]. NeurIPS, 2019.
>
> [2] Spencer J, Choudhury S, Venkatraman A, et al. Feedback in imitation learning: The three regimes of covariate shift[J]. arXiv preprint arXiv:2102.02872, 2021.

---

> > ### Author Rebuttal · Reviewer_5e6E · 2026-03-31
> >
> > Thank you for the detailed rebuttal.
> >
> > The rebuttal has addressed a substantial portion of my concerns. In particular, the authors provide additional comparisons under matched settings and include simple baselines such as random and top-k token selection. These results make it much more convincing that the performance gains are not solely due to reducing the number of tokens, but also come from the proposed adaptive sampling mechanism itself.
> >
> > The additional ablations on the predictor design are also helpful. Beyond Global Average Pooling, alternative architectures are evaluated, which strengthens the justification of the design choices.
> >
> > That said, I still have some remaining reservations. The practical inference latency improvement is relatively limited compared to the FLOPs reduction, and the explanation for the large OOD performance gain is still not fully supported by direct evidence, relying more on a plausible interpretation.
> >
> > Overall, I find that the rebuttal significantly improves the clarity and credibility of the work, and the remaining issues are not critical enough to outweigh its contributions. Therefore, I am willing to increase my score to 4 (weak accept).

---

> > > ### Author Response · Authors · 2026-04-02
> > >
> > > We sincerely thank the reviewer for engaging in the discussion, acknowledging our efforts in the rebuttal, and increasing the score to a Weak Accept.
> > >
> > > We also highly appreciate your remaining insights. You are correct that the gap between theoretical FLOPs reduction and practical wall-clock inference latency is an important nuance. Regarding the large OOD performance gains, while it is fundamentally challenging to rigorously prove the exact causal mechanism, we completely agree that more direct support is needed. To address this, we will include additional, comprehensive visualizations in the Appendix of the camera-ready version. These extended visualizations will intuitively demonstrate how the model dynamically allocates spatial tokens in the presence of unseen distractors, providing stronger qualitative evidence for our interpretation.
> > > We will also ensure that the practical latency bottlenecks and the current bounds of our OOD interpretation are transparently discussed in the Limitation section.
> > >
> > > Thank you again for your time, your constructive feedback, and for helping us make this work more rigorous and complete.

---

### Official Review · Reviewer_B6e3 · 2026-03-13

**Soundness:** 2
**Presentation:** 2
**Significance:** 3
**Originality:** 3
**Overall Recommendation:** 3
**Confidence:** 4

**Summary:**

This paper proposes GridS (Differentiable Grid Sampler), a module designed to improve the token efficiency of VLAs. Existing token pruning methods reduce computation but often degrade performance because aggressive pruning removes critical geometric details such as contact points. To address this issue, the authors reformulate visual token pruning as geometry-aware continuous resampling. GridS predicts a small set of salient coordinates using global contextual features and extracts visual tokens via differentiable bilinear interpolation, allowing the model to retain important spatial information while dramatically reducing the number of tokens. Experiments on simulation and real-world robotic tasks show that GridS can reduce visual tokens while maintaining comparable or improved success rates.

**Compliance With Llm Reviewing Policy:**

Affirmed.

**Final Justification:**

While the authors rebuttal addressed some of my concern, the authors changed the claim from learning "geometry-aware token selection", which implies a generalizable and task agonistic framework for compressing tokens, to "task-relevant spatial sampling". In my mind, this greatly change the scope of the paper and made the paper less interesting. My main concern of "task relevant" token compression is that the compression becomes uniform when the pretraining tasks are diverse, especially cross dataset or cross embodiments. Since there is no cross-datasets and cross-embodiments experiments, I retain my score to be weak reject.

**Key Questions For Authors:**

1. Figure 7 appears to show that the Information Retention map for the ALOHA dataset is nearly uniform across the image. If this is the case, does it imply that the method is effectively performing uniform downsampling rather than selecting salient regions?
2. See weakness 2.

**Limitations:**

1. The GridS sampler sampling mechanism does not incorporate language or task semantics when deciding which regions to retain. In vision-language-action settings, task instructions often specify which objects or spatial relations are relevant for the current decision. Without conditioning the sampling process on language or task context, the sampler may retain visually salient regions that are not necessarily relevant to the task. Integrating language-aware or task-aware sampling could potentially further improve efficiency by focusing tokens specifically on instruction-relevant objects.
2. The coordinate prediction mechanism relies on global average pooling of dense visual features to produce a context vector used for sampling location prediction. While efficient, this global representation may discard fine-grained spatial information that could be useful for identifying precise sampling locations. As a result, the predicted sampling coordinates may sometimes lack spatial precision, particularly in scenes with multiple interacting objects.

**Strengths And Weaknesses:**

Strengths
1. The motivation of increasing token efficiency of VLA is clear and the proposed method is able to achieve comparable performance with less tokens.
2. The grid sampler is continuous and require no additional auxiliary loss to supervise.

Weaknesses
1. There are missing ablation studies. The authors should compare with heuristic based grid sampling methods such as uniformly downsampling or sampling from the foreground. Especially when Figure 7 appears to show that the Information Retention map for the ALOHA dataset is nearly uniform across the image, which imply that the method is effectively performing uniform downsampling rather than selecting salient regions.
2. The paper claims that GridS preserves geometry-aware visual tokens, yet the training objective appears to be purely task-driven without any explicit geometric supervision or auxiliary loss encouraging geometric consistency. If there is no auxiliary loss enforcing geometric structure, how can you claim that the retained tokens are truly geometry-aware rather than simply selecting foreground or high-saliency regions? Additional analysis or visualization supporting the geometry-awareness claim would strengthen this argument.
3. It appears that each policy is trained on datasets specific to the benchmark on which it is evaluated. Since the proposed sampler is trained end-to-end with the policy, a natural concern is that the sampler may simply learn a dataset-specific foreground distribution rather than genuinely learning task-relevant geometric sampling. For example, if certain objects or spatial layouts frequently appear in a benchmark dataset, the sampler might implicitly learn to focus on those regions rather than learning a generalizable sampling strategy. To better validate the claim that GridS learns geometry-aware token selection, it would be useful to evaluate a policy trained on datasets from different benchmarks or environments, and visualize the sampled grid locations in this cross-dataset setting.

---

> ### Author Rebuttal · Authors · 2026-03-31
>
> We thank Reviewer B6e3 for the thoughtful feedback and for highlighting that our continuous grid sampler smoothly improves token efficiency without needing auxiliary losses.
>
> > **W1/Q1: Missing ablations (Uniform / Random) & Concern that Fig 7 implies uniform downsampling.**
>
> We want to clarify this mechanism and correct the understandable visual misinterpretation of Figure 7.
>
> Figure 7 visualizes feature-space information retention (max cosine similarity between original and compressed tokens), *not* spatial sampling coordinates. A "nearly uniform" map is actually the optimal outcome: it proves our 16 tokens successfully reconstruct the entire image's semantic context. Without GridS's end-to-end training, heuristic sampling yields drastically lower, sparse retention scores. This confirms GridS performs active feature enrichment, the task loss forces the network to aggregate global context into these 16 geometric bottlenecks, rather than uniformly sampling the image.
>
> To definitively prove GridS outperforms naive downsampling, we evaluated heuristic baselines on our exact $\pi_0$ architecture under the identical extreme budget (16 tokens):
>
> | Method | Tokens | Spatial | Object | Goal | Long | LIBERO Avg. SR (%) |
> | :--- | :---: | :---: | :---: | :---: | :---: | :---: |
> | Baseline | 256 | 97.2 | 98.8 | 96.0 | 85.6 | 94.4 |
> | Random (Uniform) | 16 | 93.8 | 98.4 | 85.8 | 73.2 | 87.8 |
> | Top-K Pruning | 16 | 96.6 | 98.8 | 89.8 | 77.0 | 90.5 |
> | **GridS (Ours)** | 16 | 98.0 | 99.2 | 96.4 | 90.2 | **96.0** |
>
> | Method | Tokens | TC Scripted | TC Human | Insertion Scripted | Insertion Human | ALOHA Avg. SR (%) |
> | :--- | :---: | :---: | :---: | :---: | :---: | :---: |
> | Baseline | 256 | 100.0 | 96.9 | 91.4 | 56.7 | 86.3 |
> | Random (Uniform) | 16 | 91.2 | 40.5 | 31.0 | 12.9 | 43.9 |
> | Top-K Pruning | 16 | 97.9 | 91.4 | 42.4 | 15.5 |61.8 |
> | **GridS (Ours)** | 16 | 100.0  | 96.9 | 86.9 | 64.2 | **96.0** |
>
> > TC denotes "Transfer cube" task.
>
> As shown, naive uniform downsampling (Random) and foreground heuristics (Top-K) suffer catastrophic degradation, particularly in long-horizon tasks. Only the end-to-end optimized GridS successfully retains and even surpasses the full 256-token baseline. We will add these experiment in our ablation study section.
>
> > **W2/W3: Insufficient evidence of geometric awareness without explicit loss & Dataset bias concerns.**
>
> We sincerely appreciate this constructive critique. Empirically, our "Stack Cubes" experiment (Sec. 4.2, Table 3) serves as the requested cross-environment evaluation. Tested under strict Out-of-Distribution (OOD) settings with novel layouts and unseen distractors, GridS achieved a massive +52.4% success rate improvement. New Appendix visualizations further confirm that GridS dynamically tracks critical elements rather than overfitting to static coordinates.
>
> The standard VLA paradigm relies on downstream fine-tuning, which is inherently highly task-relevant. Because GridS is jointly optimized during this stage, learning a dataset-specific, task-driven distribution is the exact intended behavior, not a structural limitation. It correctly learns to track elements (e.g., gripper, target objects) that are causally critical to the specific task at hand.
> However, we fully agree that our original term ("geometry-aware token selection") was overly broad. Following your distinction, we have revised our terminology throughout the manuscript to **"task-relevant spatial sampling."** We deeply appreciate this suggestion for improving our scientific rigor.
>
> > **L1/L2: Lack of semantic guidance & GAP precision loss.**
>
> We sincerely appreciate these insightful observations. While it is highly intuitive that GAP might lose fine-grained spatial precision and that explicit semantic guidance could improve sampling, our empirical ablations (now added to the Appendix) reveal a counter-intuitive reality in embodied manipulation:
>
> To preserve higher spatial precision, we replaced the parameter-free GAP with a parameterized Conv-Downsample. However, this actually degraded performance by **-4.4%**. The extra convolutional parameters easily overfitted to dataset-specific spatial biases.
> To explore explicit semantic guidance, we incorporated a frozen, pre-trained SAM (Segment Anything Model) to guide the coordinate sampling. Surprisingly, this resulted in a severe **-6.2%** performance drop. This demonstrates a critical divergence in robotics: generic semantic boundaries often misalign with task-driven geometric interaction points. In fact, parameter-free GAP proved to be a much more robust global context summarizer for generalizable coordinate prediction.
>
> Therefore, our current design is not a reluctant compromise, but a deliberate choice to prevent spatial overfitting and prioritize geometric interaction over pure visual semantics. We have explicitly incorporated this discussion into the revised "Limitations" section to inspire future research.

---

> > ### Author Rebuttal · Reviewer_B6e3 · 2026-04-02
> >
> > Thanks the authors for clarifying the misunderstanding of Figure 7. However, the response results in more questions. If a uniform InformationRetention map is desired, it seems in libero and real-world setting, there are areas with low InformationRetention.
> >
> > More importantly, the authors changed the claim from learning "geometry-aware token selection", which implies a generalizable and task agonistic framework for compressing tokens to "task-relevant spatial sampling". In my mind, this greatly change the scope of the paper and made the paper less interesting. If the compression is task-relevant, the paper should compare with other works selecting language related visual features based on attention maps.

---

> > > ### Author Response · Authors · 2026-04-03
> > >
> > > We sincerely thank the reviewer for the sharp observation and for pushing us to be rigorously precise with our terminology.
> > >
> > > To be absolutely clear upfront: GridS relies strictly on visual information and does not introduce any text/language information into its sampling mechanism.
> > > GridS compresses the visual tokens **before** they interact with the language conditioning in the VLA transformer. Therefore, its behavior is strictly driven by the dynamic visual scene, not the linguistic task instruction.
> > > In this context, "task-relevant" refers to identifying universal physical affordances within a scene (e.g., the end-effector and interactive objects) that are essential for execution.
> > > By dynamically sampling these geometry-critical tokens based purely on the scene's spatial content, GridS intentionally discards redundant background noise. This preserves essential physical precision, reduces computational overhead, and ensures the framework remains fully generalizable. We will refine our terminology in the revised manuscript, explicitly defining our mechanism with a full explaination to eliminate any ambiguity.
> > >
> > > Regarding the "Information Retention" map, we acknowledge that a strictly uniform map represents the theoretical "ideal" of zero information loss. However, preserving all information does not equate to the highest task success rate. In visually simple scenarios (e.g., standard ALOHA tasks) where background noise is minimal, uniform retention succeeds because a small number of tokens can easily cover the clean scene. In contrast, LIBERO and our real-world settings are heavily dominated by visual clutter. To achieve high success rates in these complex environments, the policy must intentionally discard irrelevant information. Therefore, the low-retention areas correctly reflect GridS actively filtering out background noise to concentrate capacity on critical dynamic geometries. This deliberate spatial regularization is the reason why GridS achieves a massive OOD success rate improvement over the full-token baseline in real-world settings.
> > >
> > >
> > > Furthermore, we have compared GridS against standard attention-based pruning methods (FastV, SparseVLM) on LIBERO dataset. GridS vastly outperforms them because discrete token dropping destroys continuous 2D spatial structures, whereas our resampling effectively preserves them:
> > >
> > > | Method | Tokens | Spatial | Object | Goal | Long | Avg. SR |
> > > | :--- | :---: | :---: | :---: | :---: | :---: | :---: |
> > > | $\pi_0$ | 256 | 97.2 | 98.8 | 96.0 | 85.6 | 94.4
> > > | $\pi_0$ + FastV | 100 | 97.0 | 98.4 | 93.8 | 82.4 | 92.9
> > > | $\pi_0$ + SparseVLM | 100 | 93.4 | 98.0 | 91.2 | 76.6 | 89.8
> > > | **$\pi_0$ + GridS (Ours)** | **4** | 96.6 | 99.4 | 96.4 | 89.6 | **95.5**
> > >
> > > Thank you again for helping us recognize this descriptive ambiguity. We will explicitly clarify in the final manuscript that GridS is a purely vision-driven, scene/task-relevant mechanism to ensure total accuracy.

---

### Official Review · Reviewer_LrTB · 2026-03-13

**Soundness:** 2
**Presentation:** 2
**Significance:** 2
**Originality:** 2
**Overall Recommendation:** 3
**Confidence:** 3

**Summary:**

This paper introduces GridS, a differentiable visual token pruning module for Vision-Language-Action (VLA) models that reformulates token selection as continuous, geometry-aware feature sampling. Instead of selecting discrete patches, GridS extracts features via differentiable bilinear interpolation, preserving fine-grained spatial information. Integrated as a plug-and-play component in VLA architectures (e.g., π0, π0.5), GridS compresses visual tokens by over 90% and reduces computation by up to 76% without degrading performance. Experiments on simulation and real-world robotic tasks demonstrate improved efficiency, maintained or higher success rates, and stronger robustness in out-of-distribution scenarios.

**Compliance With Llm Reviewing Policy:**

Affirmed.

**Final Justification:**

My main concern is limited ablation studies pointed by Reviewer B6e3, and limited RoboTwin 2.0 results in rebuttal.

**Key Questions For Authors:**

See Weakness

**Limitations:**

yes

**Strengths And Weaknesses:**

Strengths:
1. The proposed GridS method is well motivated and technically sound. By introducing differentiable grid sampling, it avoids the limitation of previous approaches that select discrete patches, which can lead to the loss of important target regions.
2. The paper is well written and clearly structured. The motivation, methodology, and experimental setup are presented in a logical and easy-to-follow manner, allowing readers to readily understand the core ideas and technical details.
3. The paper highlights a problem that differs from token pruning in traditional VLMs. In VLA models, it is more important to focus attention on interaction points relevant to manipulation rather than simply pruning tokens based on semantic importance.

Weakness:
1. The proposed GridS module requires training to learn the coordinate predictor, while several compared token pruning approaches (e.g., FastV, SparseVLM, VLA-Cache) are training-free. This raises a potential fairness concern in the comparison, as training-based methods may benefit from additional optimization. Moreover, the paper does not explicitly discuss the extra training cost introduced by GridS or analyze whether the efficiency gains during inference outweigh this additional training overhead.
2. Although the paper claims that GridS is a plug-and-play module applicable to various VLA architectures, the experiments are mainly conducted within a single model family (π0 and π0.5). While these results are promising, additional evaluations on more diverse architectures would strengthen the claim of general applicability.
3. Most experiments are conducted on simulation benchmarks and relatively simple manipulation tasks. The paper does not evaluate the method on more complex or long-horizon robotic tasks that involve richer interactions and longer temporal dependencies. It remains unclear whether the proposed token compression strategy would maintain its effectiveness in more challenging settings (e.g., complex multi-stage manipulation benchmarks such as RoboTwin2.0[1] environments).
4. Although the method predicts task-aware sampling coordinates, the paper provides limited analysis of where these sampling points are actually placed during execution. It is therefore difficult to assess whether the model consistently focuses on interaction-critical regions (e.g., contact points or tool-object boundaries).

[1] Chen T, Chen Z, Chen B, et al. Robotwin 2.0: A scalable data generator and benchmark with strong domain randomization for robust bimanual robotic manipulation[J]. arXiv preprint arXiv:2506.18088, 2025.

---

> ### Author Rebuttal · Authors · 2026-03-31
>
> We sincerely thank Reviewer LrTB for recognizing that GridS is well-motivated, structurally clear, and successfully addresses the limitations of discrete token pruning.
>
> > **W1: Fairness compared to training-free methods & Training cost overhead.**
>
> We respectfully clarify that the "training-free" nature of prior pruning methods (e.g., FastV, SparseVLM) is not an inherent advantage, but a forced compromise due to their non-differentiable designs, which inherently causes downstream performance degradation.
>
> Unlike existing training-free pruning methods that are applied post-training and inevitably cause downstream performance degradation (-0.2% to -2.0%), GridS is fully differentiable and integrates directly into the standard VLA fine-tuning stage. By jointly optimizing the pruning mechanism with the task loss, GridS effectively eliminates the performance drops associated with non-differentiable methods. Furthermore, rather than introducing overhead, this integration drastically reduces fine-tuning costs, achieving a 3.4$\times$ training speedup for $\pi_0$.
>
> To ensure a strict comparison, we evaluated these methods on the exact same $\pi_0$ base architecture and 30k-iteration fine-tuning protocol:
>
> |Method|Tokens|Spatial|Object|Goal|Long|Avg. SR|
> |:---|:---:|:---:|:---:|:---:|:---:|:---:|
> |Baseline|256|97.2|98.8|96.0|85.6|94.4
> |FastV|100|97.0|98.4|93.8|82.4|92.9
> |SparseVLM|100|93.4|98.0|91.2|76.6|89.8
> |**GridS (Ours)**|**4**|96.6|99.4|96.4|89.6|**95.5**
>
> As shown above, non-differentiable methods suffer significant performance drops even when retaining 100 tokens. In stark contrast, GridS achieves superior accuracy using only 4 tokens. This proves that GridS does not merely drop tokens, but actively learns to sample the most geometry-critical regions. We will include this comparison in the revised manuscript.
>
> > **W2: Evaluation on more diverse architectures.**
>
> We respectfully point out that our evaluation extends well beyond the $\pi_0$ family. In our original manuscript, the real-world stacking experiments were successfully conducted using SmolVLA, a distinctly different architecture. Furthermore, to explicitly strengthen our claim of general applicability, we have newly integrated GridS with the XVLA architecture for complex, long-horizon tasks on the RoboTwin 2.0 benchmark(mentioned in next paragraph "W3"). The seamless integration and significant performance gains across these four diverse model families ($\pi_0$, $\pi_{0.5}$, SmolVLA, and XVLA) firmly validate that GridS is a highly generalizable, plug-and-play module. We will emphasize this architectural diversity more prominently in the revision.
>
>
> > **W3: Lack of evaluation on complex or long-horizon manipulation tasks (e.g., RoboTwin 2.0).**
>
> We thank the reviewer for the constructive suggestion to evaluate our method on complex, long-horizon tasks. To directly address this, we have expanded our evaluation in two key areas to demonstrate GridS's effectiveness in extended temporal settings.
>
> We conducted new experiments on the complex "Place Bread Skillet" task. This suite is explicitly designed to evaluate long-horizon planning and sequential multi-stage execution. As shown in our new results, GridS effectively maintains geometric attention over extended horizons, vastly outperforming the full-token baseline: while the fine-tuned standard XVLA [1] achieves only a 12% success rate, our XVLA+GridS policy yields a remarkable 76% success rate. We will add more result of RoboTwin 2.0 on the next period of discussion.
> Furthermore, our existing real-world stacking task is inherently a multi-stage process (approach $\rightarrow$ grasp $\rightarrow$ align $\rightarrow$ stack) requiring sustained spatial reasoning. GridS achieved a massive +52.4% success rate improvement here over the baseline.
>
> Together, these results clearly demonstrate that GridS does not degrade in challenging settings; rather, it robustly supports rich, long-horizon interactions by preventing spatial error accumulation. We will include these new RoboTwin 2.0 results in the revised manuscript.
>
> > **W4: Analysis of the actual spatial distribution of sampled points.**
>
> We fully agree that visual transparency is crucial. We have added a "Spatial Tracking Analysis" section with step-by-step visualizations to the revised Appendix.
> These new visualizations explicitly confirm your intuition: the predicted GridS coordinates densely cluster on exactly the interaction-critical regions you highlighted. Furthermore, they dynamically track these contact points as the robot moves.
> This explicit visual evidence of tool, object tracking explains why GridS achieves the massive +52.4% real-world OOD improvement, it successfully ignores background distractors and forces the token budget entirely onto causal geometric features.
>
> [1] Zheng J, Li J, Wang Z, et al. X-vla: Soft-prompted transformer as scalable cross-embodiment vision-language-action model[J]. arXiv preprint arXiv:2510.10274, 2025.

---

> > ### Author Rebuttal · Reviewer_LrTB · 2026-04-03
> >
> > After reading rebuttal and comments from other reviewers, I choose to keep my initial rating.

---

### Decision · Program_Chairs · 2026-04-30

**Decision:**

Accept (regular)

**Comment:**

GridS introduces a differentiable, plug-and-play framework for efficient robotic control, enabling significant visual token compression through geometry-aware feature sampling. The paper has received mixed reviews. While the authors provided substantial new evidence regarding experimental fairness, training overhead, and architectural generalizability that satisfied several reviewers, some conceptual concerns regarding the broader generalizability of the sampling strategy remain. Given the technical clarifications and the performance gains on complex benchmarks such as RoboTwin 2.0, the paper is recommended for a weak acceptance if space permits.